# Most cancers carry a substantial deleterious load due to Hill-Robertson interference

Susanne Tilk[1]*, Svyatoslav Tkachenko[2], Christina Curtis[3,4,5], Dmitri A Petrov[1], Christopher D McFarland[2]*

[1]Department of Biology, Stanford University, Stanford, United States; [2]Department of Genetics and Genome Sciences, Case Western Reserve University, Cleveland, United States; [3]Department of Medicine, Division of Oncology, Stanford University School of Medicine, Stanford, United States; [4]Department of Genetics, Stanford University School of Medicine, Stanford, United States; [5]Stanford Cancer Institute, Stanford University School of Medicine, Stanford, United States

**Abstract** Cancer genomes exhibit surprisingly weak signatures of negative selection (Martincorena et al., 2017; Weghorn, 2017). This may be because selective pressures are relaxed or because genome-wide linkage prevents deleterious mutations from being removed (Hill-Robertson interference; Hill and Robertson, 1966). By stratifying tumors by their genome-wide mutational burden, we observe negative selection ($dN/dS$ ~ 0.56) in low mutational burden tumors, while remaining cancers exhibit $dN/dS$ ratios ~1. This suggests that most tumors do not remove deleterious passengers. To buffer against deleterious passengers, tumors upregulate heat shock pathways as their mutational burden increases. Finally, evolutionary modeling finds that Hill-Robertson interference alone can reproduce patterns of attenuated selection and estimates the total fitness cost of passengers to be 46% per cell on average. Collectively, our findings suggest that the lack of observed negative selection in most tumors is not due to relaxed selective pressures, but rather the inability of selection to remove deleterious mutations in the presence of genome-wide linkage.

*For correspondence:
tilk@stanford.edu (ST);
cdm113@case.edu (CDMcF)

**Competing interest:** The authors declare that no competing interests exist.

## Editor's evaluation

This is an important paper that shows most cancers unavoidably accumulate damaging mutations. Whilst the majority of claims are convincingly supported by the data, evidence that damaging changes are buffered by heat shock pathways is currently incomplete. The insights into selection efficiency are important for the understanding of cancer growth and response to therapy. A broader implication is that high mutation load tumors may use common strategies to tolerate accumulated deleterious mutations, providing a therapeutic target.

## Introduction

Tumor progression is an evolutionary process acting on somatic cells within the body. These cells acquire mutations over time that can alter cellular fitness by either increasing or decreasing the rates of cell division and/or cell death. Mutations which increase cellular fitness (drivers) are observed in cancer genomes more frequently because natural selection enriches their prevalence within the tumor population (*Martincorena et al., 2017*; *Weghorn and Sunyaev, 2017*). This increased prevalence of mutations across patients within specific genes is used to identify driver genes. Conversely, mutations that decrease cellular fitness (deleterious passengers) are expected to be observed less frequently.

This enrichment or depletion is often measured by comparing the expected rate of nonsynonymous mutations (*dN*) accruing within a region of the genome to the expected rate of synonymous mutations (*dS*), which are presumed to be neutral. This ratio, *dN/dS*, is expected to be below 1 when the majority of nonsynonymous mutations are deleterious and removed by natural selection, be ~1 when all nonsynonymous mutations are neutral, and can be >1 when a substantial proportion of nonsynonymous mutations are advantageous.

Two recent analyses of *dN/dS* patterns in cancer genomes found that for most nondriver genes *dN/dS* is ~1 and that only 0.1–0.4% of genes exhibit detectable negative selection (*dN/dS* < 1) (**Martincorena et al., 2017**; **Weghorn and Sunyaev, 2017**).This differs substantially from patterns in human germline evolution where most genes show signatures of negative selection (*dN/dS* ~ 0.4) (**Martincorena et al., 2017**). Two explanations for this difference have been posited. First, the vast majority of nonsynonymous mutations may not be deleterious in somatic cellular evolution despite their deleterious effects on the organism. While most genes may be critical for proper organismal development and multicellular functioning, they may not be essential for clonal tumor growth. In this hypothesis, negative selection (*dN/dS* < 1) should be observed only within essential genes and absent elsewhere (*dN/dS* ~ 1). While appealing in principle, most germline selection against nonsynonymous variants appears to be driven by protein misfolding toxicity (**Drummond and Wilke, 2008**; **Lobkovsky et al., 2010**), in addition to gene essentiality. These damaging folding effects ought to persist in somatic evolution.

A second hypothesis is that even though many nonsynonymous mutations are deleterious in somatic cells, natural selection fails to remove them. One possible reason for this inefficiency is the unique challenge of evolving without recombination. Unlike sexually recombining germline evolution, tumors must evolve under genome-wide linkage that creates interference between mutations, known as Hill-Robertson interference, which reduces the efficiency of natural selection (**Hill and Robertson, 1966**). Without recombination to link and unlink combinations of mutations, natural selection must act on entire genomes – not individual mutations – and select for clones with combinations of mutations of better aggregate fitness. Thus, advantageous drivers may not fix in the population, if they arise on an unfit background, and conversely, deleterious passengers can fix, if they arise on fit backgrounds.

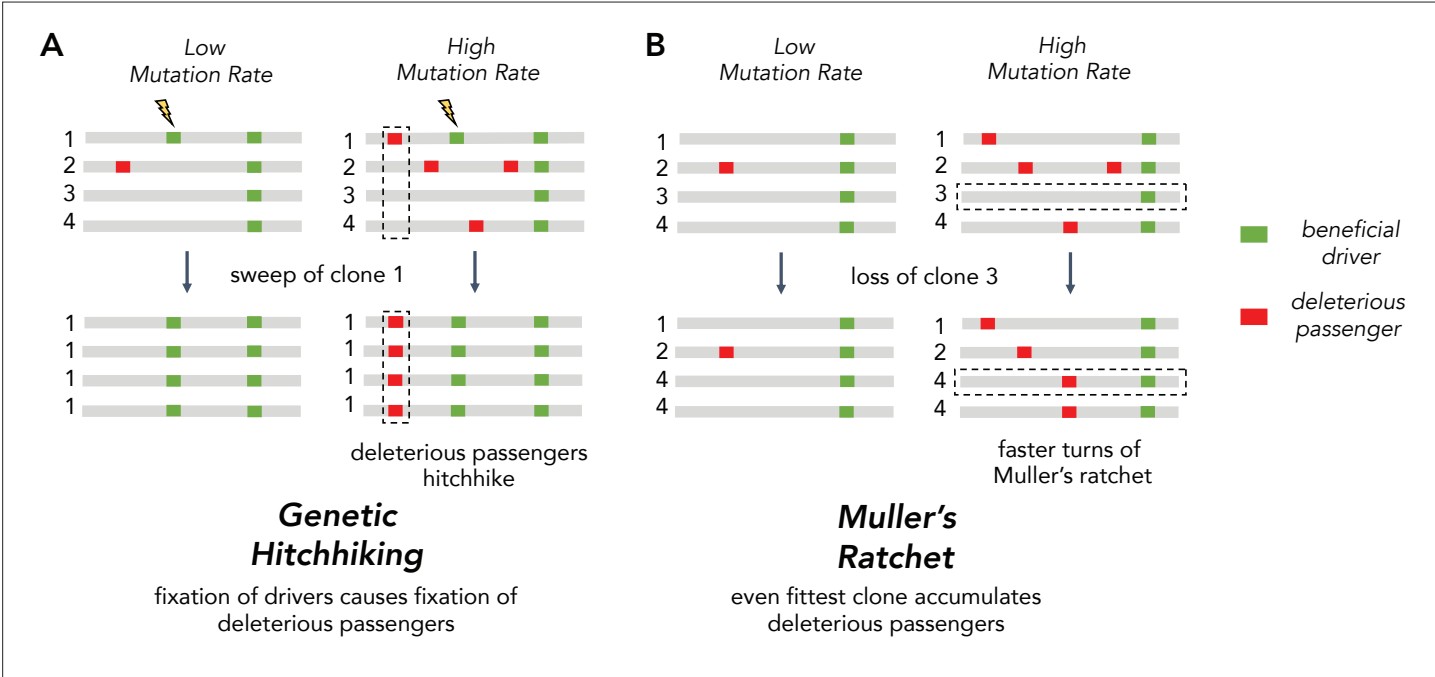

**Figure 1.** Two Hill-Robertson interference processes that accumulate deleterious mutations at high mutation rates. (**A**) Genetic hitchhiking. Each number identifies a different segment of a clone genome within a tumor. De novo beneficial driver mutations that arise in a clone can drive other mutations (passengers) in the clone to high frequencies (black dotted column). If the passenger is deleterious, both beneficial drivers and deleterious passengers can accumulate. (**B**) Muller's ratchet. As the mutation rate within a tumor increases, deleterious passengers accumulate on more clones. If the fittest clone within the tumor is lost through genetic drift (black dotted row), the overall fitness of the population will decline.

The inability of asexuals to eliminate deleterious passengers is driven by two Hill-Robertson interference processes: *hitchhiking* and *Muller's ratchet* (*Figure 1A*). Hitchhiking occurs when a strong driver arises within a clone already harboring several passengers. Because these passengers cannot be unlinked from the driver under selection, they are carried with the driver to a greater frequency in the population. Muller's ratchet is a process where deleterious mutations continually accrue within different clones in the population until natural selection is overwhelmed. Whenever the fittest clone in an asexual population is lost through genetic drift, the maximum fitness of the population declines to the next most fit clone (*Figure 1B*). The rate of hitchhiking and Muller's ratchet both increase with the genome-wide mutation rate (*Johnson, 1999*; *Neher and Shraiman, 2012*). Therefore, the second hypothesis predicts that selection against deleterious passengers should be more efficient (*dN/dS* < 1) in tumors with lower mutational burdens.

Here, we leverage the 10,000-fold variation in tumor mutational burden across 33 cancer types to quantify the extent that selection attenuates, and thus becomes more inefficient, as the mutational burden increases. Using *dN/dS*, we find that selection against deleterious passengers and in favor of advantageous drivers is most efficient in low mutational burden cancers. Furthermore, low mutational burden cancers exhibit efficient selection across cancer subtypes, as well as within subclonal mutations, homozygous mutations, somatic copy number alterations (CNAs), and essential genes. Additionally, high mutational burden tumors appear to mitigate this deleterious load by upregulating protein folding and degradation machinery. Finally, using evolutionary modeling, we find that Hill-Robertson interference alone can in principle explain these observed patterns of selection. Modeling predicts that most cancers carry a substantial deleterious burden (~46%) that necessitates the acquisition of multiple strong drivers (~5) in malignancies that together provide a benefit of ~119%. Collectively, these results explain why signatures of selection are largely absent in cancers with elevated mutational burdens and indicate that the vast majority of tumors harbor a large mutational load.

## Results

### Null models of mutagenesis in cancer

Mutational processes in cancer are heterogeneous, which can bias *dN/dS* estimates of selective pressures. *dN/dS* overcomes this issue by dividing observed mutation counts by what is expected under neutral evolution using null models. These null models must account for mutational biases that are often specific to cancer types and genomic regions.

To ensure our *dN/dS* calculations are robust and reproducible, we applied two different methods to account for mutational biases. The first approach uses a previously established parametric mutational model (*dNdScv*) that explicitly estimates the background mutational bias of each gene in its calculation of *dN/dS* (*Martincorena et al., 2017*). The second approach uses a permutation-based, non-parametric (parameter-free) estimation of *dN/dS*. In this approach, every observed mutation is permuted while preserving the gene, patient samples, specific base change (e.g. A>T) and its trinucleotide context. Note that permutations do not preserve the codon position of a mutation and thus can change its protein coding effect (nonsynonymous vs. synonymous). The permutations are then tallied for both nonsynonymous $d_N^{(permuted)}$ and synonymous $d_S^{(permuted)}$ substitutions (*Figure 2— figure supplement 1*) and used as expected proportional values for the observed number of nonsynonymous $d_N^{(observed)}$ (or simply $d_N$) and synonymous $d_S^{(observed)}$ ($d_S$) mutations in the absence of selection. The unbiased effects of selection on a gene, *dN/dS*, is then:

$$\frac{dN}{dS} = \frac{d_N^{(observed)}/d_N^{(permuted)}}{d_S^{(observed)}/d_S^{(permuted)}}$$

For all cancer types and patient samples, p-values and confidence intervals are determined by bootstrapping patient samples. Note that this permutation procedure will account for gene and tumor-level mutational biases (e.g. neighboring bases [*Alexandrov and Stratton, 2014*], transcription-coupled repair, S phase timing [*Haradhvala et al., 2016*], mutator phenotypes) and their covariation. We confirmed that this approach accurately measures selection even in the presence of simulated mutational biases (Materials and methods, *Figure 2—figure supplement 2A*). In addition, this approach also reliably measures the absence of selection (*dN/dS* = 1) in weakly expressed genes (*Figure 2— figure supplement 2C*).

We find that both the parametric and non-parametric approaches identify similar patterns of selection (*Figure 2A*). Since parametric mutational models can become very complex in cancer (exceeding 5000 parameters in some cases; *Martincorena et al., 2017*; *Zapata et al., 2018*), we elected to use the non-parametric approach, which makes fewer assumptions about underlying mutational processes, in subsequent calculations of *dN/dS*.

## Attenuation of selection in drivers and passengers for elevated mutational burden tumors

We estimated *dN/dS* patterns in both driver and passenger gene sets across 10,288 tumors from TCGA aggregated over 33 cancer types (*Ellrott et al., 2018*) (Materials and methods). Since TCGA is composed of whole-exome data, which limits our ability to assess mutations in non-coding regions, we elected to use the total number of protein-coding mutations as our proxy for the mutational burden of tumors. To quantify the extent that selection attenuates as the mutational burden increases, we stratified tumors into bins based on their total number of substitutions on a log-scale. For each bin of tumors, we pooled all of the variants together and estimated *dN/dS* jointly. Consistent with the inefficient selection model, whereby selection fails to eliminate deleterious mutations in high mutational burden tumors, we observe pervasive selection against passengers exclusively in tumors with low mutational burdens (*dN/dS* ~ 0.56 in tumors with ≤3 substitutions, while *dN/dS* ~ 0.93 in tumors with >10 substitutions, *Figure 2A*). We observed little negative selection in passenger genes when aggregating tumors across all mutational burdens (*dN/dS* ~ 0.93), which is broadly similar to previous estimates (*Martincorena et al., 2017*; *Weghorn and Sunyaev, 2017*; *Zapata et al., 2018*; *Ostrow et al., 2014*).

We confirmed that negative selection on passengers is specific to low mutational burden tumors and not biased by small sample sizes (*Figure 2—figure supplement 2B*). We randomly sampled passengers from high mutational burden tumors (>10 substitutions) 1000 times using the same bin sizes in *Figure 2A* and calculated *dN/dS*. Within the smallest bin size (*N*=168 somatic nucleotide variant [SNVs]), negative selection on passengers sampled from high mutational burden tumors was absent (average *dN/dS* ~ 0.96) compared to observed *dN/dS* in low mutational burden tumors (*dN/dS* ~ 0.56; $p<2.2^{-16}$). In fact, only 1.7% of randomly sampled sets of sites had similar signals of negative selection (*dN/dS* < 0.56).

Also consistent with the inefficient selection model, drivers exhibit a similar but opposing trend of attenuated selection at elevated mutational burdens (*dN/dS* ~ 2.7 when in tumors with ≤3 substitutions and *dN/dS* gradually declines to ~1.16 in tumors with >100 substitutions). This pattern is not specific to drivers that are oncogenes or tumor suppressors (*Figure 2—figure supplement 3*). While the attenuation of selection against passengers in higher mutational burden tumors is a novel discovery, this pattern among drivers has been reported previously (*Martincorena et al., 2017*). Furthermore, we confirmed that these patterns are robust to the choices that we made in our analysis pipeline. These include the: (i) effects of germline SNP contamination (*Figure 2—figure supplement 4*), (ii) choice of driver gene set (*Bailey et al., 2018*, IntOGen *Gonzalez-Perez et al., 2013*, and COSMIC *Tate et al., 2019*; *Forbes et al., 2008*, *Figure 2—figure supplement 5*), (iii) differences in tumor purity and thresholding (*Figure 2—figure supplement 6*), and (iv) null model of mutagenesis (*dNdScv*, *Figure 2A* and *Figure 2—figure supplement 7*; *Martincorena et al., 2017*) (Materials and methods).

If negative selection is more pronounced in low mutational burden tumors, then the nonsynonymous mutations observed should also be less functionally consequential. By annotating the functional effect of all missense mutations using PolyPhen2 (*Adzhubei et al., 2010*; *Figure 2B*), we indeed find that observed nonsynonymous passengers are less damaging in low mutational burden cancers. Similarly, driver mutations become less functionally consequential as mutational burden increases, as expected for mutations experiencing inefficient positive selection (*Figure 2B*). Together these two trends provide additional and orthogonal evidence that selective forces on nonsynonymous mutations are more efficient in low mutational burden cancers.

Since all mutational types experience Hill-Robertson interference, attenuated selection should also persist in CNAs. We used two previously published statistics to quantify selection in CNAs: breakpoint frequency (*Korbel et al., 2007*) and fractional overlap (*Zack et al., 2013*). For both measures, we compare the number of CNAs that either terminate (breakpoint frequency) within or partially overlap

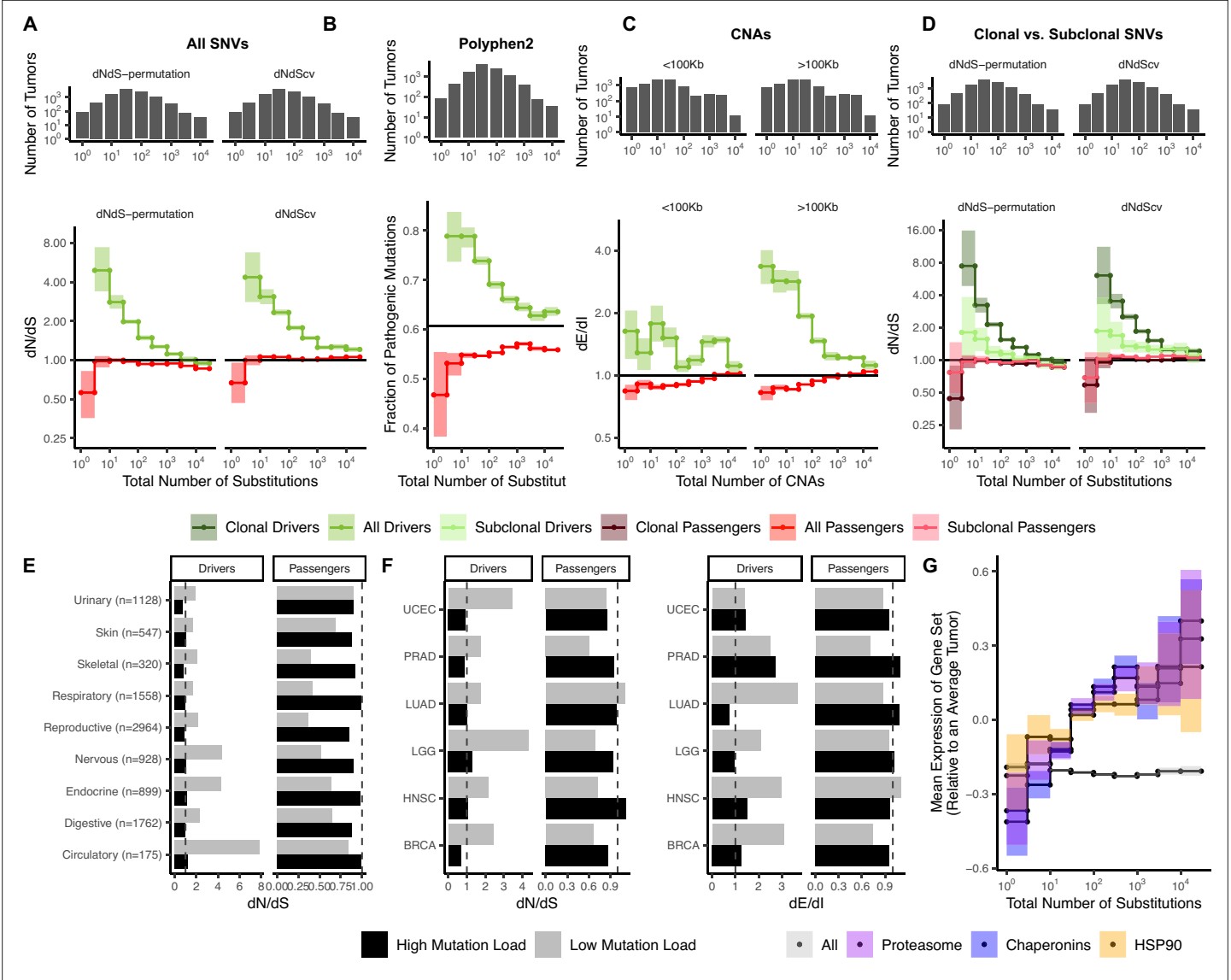

**Figure 2.** Attenuation of selection and increased protein folding stress in high mutation load tumors. (**A**) *dN/dS* of passenger (red) and driver (green) gene sets within 10,288 tumors in TCGA stratified by total number of substitutions present in the tumor ($d_N^{(observed)} + d_S^{(observed)}$). *dN/dS* is calculated with error bars using a permutation-based null model (left) and *dNdScv* (right). A *dN/dS* of 1 (solid black line) is expected under neutrality. Solid gray line denotes pan-cancer genome-wide *dN/dS*. (**B**) Fraction of pathogenic missense mutations, annotated by PolyPhen2, in the same driver and passenger gene sets also stratified by total number of substitutions. Black line denotes the pathogenic fraction of missense mutations across the entire human genome. (**C**) Breakpoint frequency of copy number alterations (CNAs) that reside within exonic (*dE*) to intergenic (*dI*) regions within putative driver and passenger gene sets (identified by GISTIC 2.0, Materials and methods) in tumors stratified by the total number of CNAs present in each tumor and separated by CNA length. Solid black line of 1 denotes values expected under neutrality. (**D**) *dN/dS* of clonal (variant allele frequency [VAF] > 0.2; darker colors) and subclonal (VAF < 0.2; lighter colors) passenger and driver gene sets in tumors stratified by the total number of substitutions. A *dN/dS* of 1 (solid black line) is expected under neutrality. (**A–D**) Histogram counts of tumors within mutational burden bins are shown in the top panels. (**E**) Driver and passenger *dN/dS* values of the highest and lowest defined mutational burden bin in broad anatomical sub-categories. (**F**) Same as (**E**), except for all specific cancer subtypes with ≥500 samples. (**G**) Z-scores of median gene expression within all genes, HSP90, Chaperonin, and Proteasome gene sets averaged across patients (relative to an average tumor) stratified by the total number of substitutions. All shaded error bars are 95% confidence intervals determined by bootstrap sampling.

The online version of this article includes the following figure supplement(s) for figure 2:

**Figure supplement 1.** Schematic of our permuted *dN* and *dS* calculation.

**Figure supplement 2.** Permutation-based null model of mutagenesis corrects for mutational biases in *dN/dS* calculations.

**Figure supplement 3.** Attenuation of selection with increasing mutational burden in both oncogenes and tumor suppressors.

*Figure 2 continued on next page*

*Figure 2 continued*

**Figure supplement 4.** No common germline polymorphisms observed in low mutation rate cancers.

**Figure supplement 5.** Weaker signals of positive selection within cancer-specific drivers.

**Figure supplement 6.** Patterns of attenuated selection persist across tumor purity thresholds.

**Figure supplement 7.** Comparison of *dN/dS* to results in *Martincorena et al., 2017*, for tumors stratified by mutational burden.

**Figure supplement 8.** Random permutations of the positions of observed copy number alterations (CNAs) exhibit neutral values of *dE/dI*.

**Figure supplement 9.** Fractional overlap of copy number alterations (CNAs) within exomic regions (*dE*) relative to intergenic regions (*dI*) exhibits similar patterns of selection as fractional overlap.

**Figure supplement 10.** Signal of negative selection in subclonal mutations are robust to variant allele frequency (VAF) threshold.

**Figure supplement 11.** Attenuation of negative selection within different functional gene sets.

**Figure supplement 12.** Attenuation of selection in somatic nucleotide variants (SNVs) persists across cancer subtypes and broad cancer group categories.

**Figure supplement 13.** Attenuation of selection in copy number alterations (CNAs) in cancer subtypes and broad cancer group categories.

**Figure supplement 14.** Upregulation of heat shock protein pathways in tumors with elevated mutational burdens.

**Figure supplement 15.** The power to detect signals of selection is dependent on the quality of mutation calls.

**Figure supplement 16.** Quantity of mutations within each mutational burden bin for data depicted in *Figure 2*.

(fractional overlap) **E**xonic regions of the genome relative to non-coding (**I**ntergenic and **I**ntronic) regions (*dE/dI*, see Materials and methods). Like *dN/dS*, *dE/dI* is expected to be <1 in genomic regions experiencing negative selection, >1 in regions experiencing positive selection (e.g. driver genes), and ~1 when selection is absent or inefficient (*Figure 2—figure supplement 8*). Using *dE/dI*, we observe attenuating selection in both driver and passenger CNAs as the total number of CNAs increases for both breakpoint frequency (*Figure 2C*) and fractional overlap (*Figure 2—figure supplement 9*). While CNAs of all lengths experience attenuated selection, CNAs longer than the average gene length (>100 KB) experience greater selective pressures in drivers. Collectively, these results strongly support the inefficient selection model and argue that the observed patterns must be due to a universal force in tumor evolution. We find that selection consistently attenuates in both drivers and passengers across all cancers as mutational burden increases.

## Strong selection in low mutational burden tumors cannot be explained by mutational timing, gene function, or tumor type

We next tested alternative hypotheses to the inefficient selection model. We considered the possibility that selection is strong only during normal tissue development, but absent after cells have transformed to malignancy. This would disproportionately affect low mutational burden tumors, as a greater proportion of their mutations arise prior to tumor transformation. If true, then attenuated selection should be absent in subclonal mutations, which must arise during tumor growth. However, selection clearly attenuates with increasing mutational burden for the subset of likely subclonal mutations with variant allele frequency (VAF) below 20% (*Figure 2D* and *Figure 2—figure supplement 10*). Although selection attenuates in drivers and passengers in both subclonal and clonal mutations, selection is weaker in both drivers and passengers with lower VAFs. Weaker efficiency of selection among less frequent variants is expected under a range of population genetic models (*Messer, 2009*) and especially so in rapidly expanding, spatially constrained cancers (*Sottoriva et al., 2015*). In addition, heterozygous mutations, to the extent they are only partially dominant (*López et al., 2020*), are also expected to exhibit lower VAFs and experience weaker selection.

Next, we considered and rejected the possibility that attenuated selection is limited to particular types of genes. We first annotated our observed mutations by different functional categories and Gene Ontology (GO) terms (*Harris et al., 2004*) and find that negative selection is not specific to any particular gene functional category expected to be under constraint, and specifically not limited to essential or housekeeping genes – a key prediction of the 'weak selection' model (*Martincorena et al., 2017*; *Figure 2—figure supplement 11*, p<0.05, Wilcoxon signed-rank test).

Finally, we found that these patterns of attenuated selection persist across cancer subtypes for both SNVs and CNAs. We calculated *dN/dS* in tumors grouped by nine broad anatomical sub-categories (e.g. neuronal) and 33 subtype classifications (*Grossman et al., 2016*; *Figure 2E–F*). We find that

patterns of attenuated selection in SNVs persists in the broad and specific (drivers $p=3.8 \times 10^{-5}$, passengers $p=1.7 \times 10^{-2}$, Wilcoxon signed-rank test; *Figure 2—figure supplement 12*) classification schemes. Furthermore, *dE/dI* measurements of CNAs exhibit similar patterns of selection in broad (*Figure 2—figure supplement 13*) and specific subtypes (*Figure 2F*; drivers p<0.05 and passengers p<0.05).

Collectively, these results suggest that tumors with elevated mutational burdens carry a substantial deleterious load. Since nonsynonymous mutations are thought to be primarily deleterious by inducing protein misfolding (*Drummond and Wilke, 2008*; *Lobkovsky et al., 2010*), we tested whether an increase in the number of passenger mutations in tumors would lead to elevated protein folding stress, and, in turn, drive the upregulation of heat shock and protein degradation (*McGrail et al., 2020*) pathways in cancer (*Santagata et al., 2011*). Indeed, gene expression of HSP90, Chaperonins, and the Proteasome does increase across the whole range of SNV (weighted $R^2$ of 0.84, 0.78, and 0.78, respectively) and CNA burdens (weighted $R^2$ of 0.83, 0.88, and 0.85, respectively) (*Figure 2G* and *Figure 2—figure supplement 14A*). This trend persists across cancer types for SNVs and CNAs (*Figure 2—figure supplement 14D-E*). Importantly, expression of these gene sets increases across the whole range of mutational burdens, even after the *dN/dS* of passengers approaches 1. This result presents additional evidence that passengers continue to impart a substantial cost to cancer cells, even in high mutational burden tumors.

## Evolutionary modeling estimates the fitness effects of drivers and passengers, and rate of Hill-Robertson interference processes

We next tested whether Hill-Robertson interference – a process where selection becomes inefficient due to interference between linked mutations with competing fitness effects – alone can generate these patterns of attenuated selection. Specifically, we modeled tumor progression as a simple evolutionary process with advantageous drivers and deleterious passengers. We then used approximate Bayesian computation (ABC) to compare these simulations to observed data and infer the mean fitness effects of drivers and passengers.

Our previously developed evolutionary simulations model a well-mixed population of tumor cells that can randomly acquire advantageous drivers and deleterious passengers during cell division (*McFarland et al., 2013*). The product of the individual fitness effects of these mutations determines the relative birth and death rate of each cell, which in turn dictates the population size $N$ of the tumor. If the population size of a tumor progresses to malignancy ($N$>1,000,000) within a human lifetime (≤100 years), the accrued mutations and patient age are recorded. The mutation rate of each simulated tumor is randomly sampled from a broad range ($10^{-12}$–$10^{-7}$ mutations · nucleotide$^{-1}$ · generation$^{-1}$, Materials and methods). Although this model ignores a great deal of known tumor biology, we believe it constitutes the simplest evolutionary model that could possibly recapitulate observed selection for drivers and against passengers. Our question is not whether this model is correct in all details but rather whether even such a simple model can generate quantitatively similar patterns as observed in the data with sensible values of mutation rates and selection coefficients.

*Figure 3A* illustrates the ABC procedure. To compare our model to observed data, we simulated an exponential distribution of fitness effects (DFEs) with mean fitness values that spanned a broad range ($10^{-2}$–$10^{0}$ for driver and $10^{-4}$–$10^{-2}$ for passengers, Materials and methods). We summarized observed and simulated data using statistics that capture three relationships: (i) the dependence of driver and passenger *dN/dS* rates on mutational burden, (ii) the rate of cancer age incidence (SEERs database *National Cancer Institute, 2007*), and (iii) the distribution of mutational burdens (summary statistics of (ii) and (iii) were based on theoretical parametric models *Frank, 2007*, Materials and methods, *Figure 3—figure supplements 1–2*). We then inferred the posterior probability distribution of mean driver fitness benefit and mean passenger fitness cost using a rejection algorithm that we validated using leave-one-out cross validation (CV) (Materials and methods, *Figure 3—figure supplement 3*).

Using this approach, the maximum likelihood estimate (MLE) of mean driver fitness benefit is 53% (*Figure 3B*), while the MLE of passenger mean fitness cost is 1.03% (*Figure 3C*). Simulations with these MLE values agree well with all observed data (*Figure 3D–F*, Pearson's *r*=0.988 for combined driver/passenger *dN/dS*).

While Hill-Robertson interference alone explains *dN/dS* rates in the passengers well, the simulations most consistent with observed data still exhibited consistently higher *dN/dS* rates in drivers

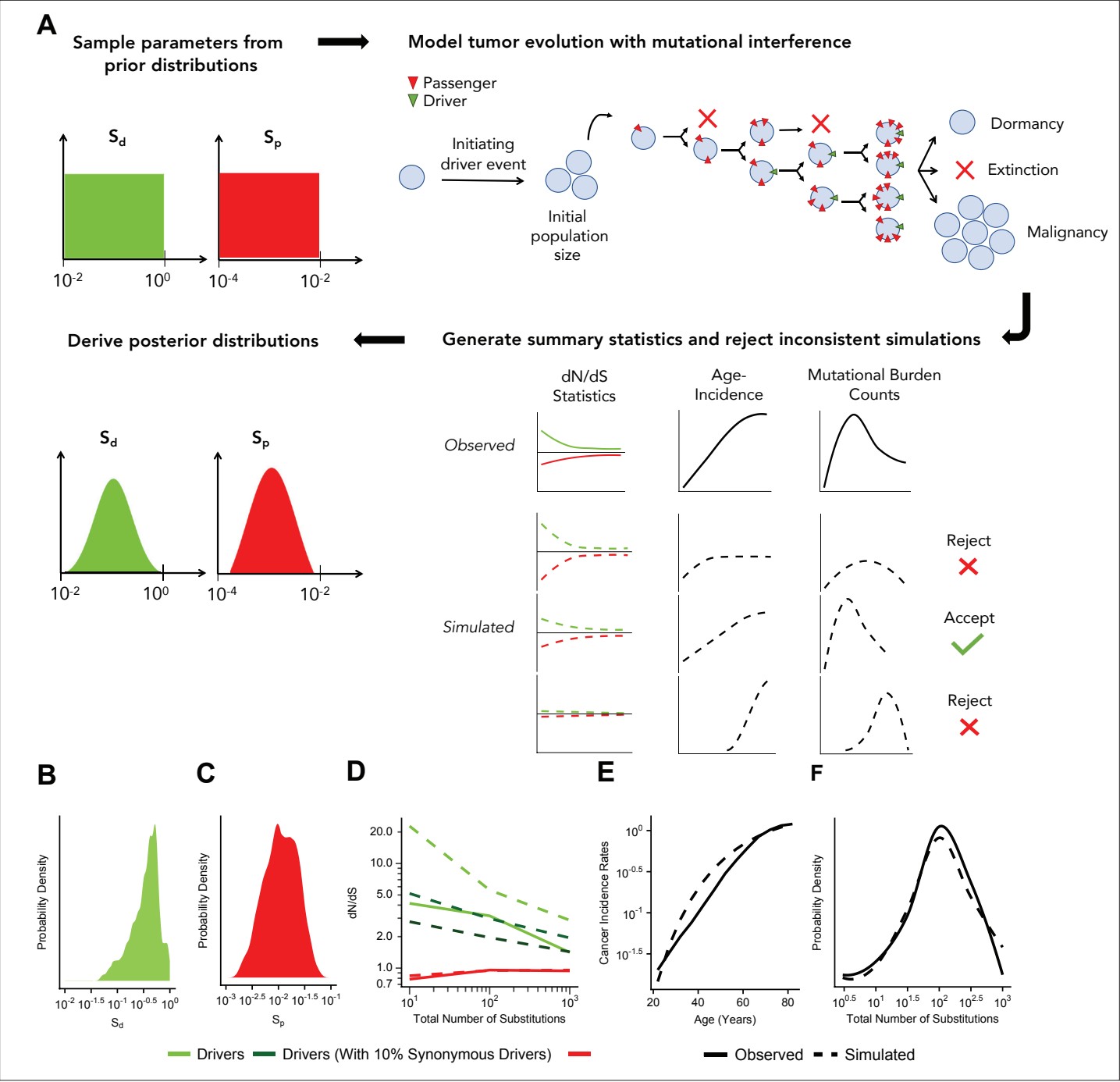

**Figure 3.** Approximate Bayesian computation (ABC) procedure estimates the strength of selection in passengers and drivers. (**A**) Schematic overview of the ABC procedure used. A model of tumor evolution with genome-wide linkage contains two parameters – $s_{drivers}$ (mean fitness benefit of drivers) and $s_{passengers}$ (mean fitness cost of passengers) – sampled over broad prior distributions of values. Simulations begin with an initiating driver event that establishes the initial population size of the tumor. The birth rate of each individual cell within the tumor is determined by the total accumulated fitness effects of drivers and passengers. If the final population size of the tumor exceeds 1 million cells within a human lifetime (100 years), patient age and accrued mutations are recorded. Summary statistics of four relationships are used to compare simulations to observed data: (**i**) $dN/dS$ rates of drivers and (**ii**) passengers across mutational burden, (**iii**) rates of cancer incidence vs. age, and (**iv**) the distribution of mutational burdens. Simulations that excessively deviate from observed data are rejected (Materials and methods). (**B–C**) Inferred posterior probability distributions of $s_{drivers}$ and $s_{passengers}$. The maximum likelihood estimate (MLE) of $s_{drivers}$ is 53.0% (green, 95% CI [16.0, 111.4]), and the MLE of $s_{passengers}$ is 1.03% (green, 95% CI [0.40, 3.98%]). (**D–F**) Comparison of the summary statistics of the best-fitting simulations (MLE parameters, dashed lines) to observed data (solid lines). (**D**) $dN/dS$ rates of passengers (red) and drivers (light green) for simulated and observed data vs. mutational burden. A model where 6% of synonymous mutations within

*Figure 3 continued on next page*

*Figure 3 continued*

drivers experience positive selection (dark green) was also considered. (**E**) Cancer incidence rates for patients above 20 years of age. (**F**) Distribution of the mutational burdens of tumors.

The online version of this article includes the following figure supplement(s) for figure 3:

**Figure supplement 1.** *dN/dS* rates of drivers and passengers in simulated cancers with various fitness coefficients.

**Figure supplement 2.** Probability of cancer by age and mutational burdens in simulated cancers at various fitness coefficients.

**Figure supplement 3.** Implementation and use of approximate Bayesian computation (ABC) for model selection and parameter estimation.

**Figure supplement 4.** Evidence of positive selection on synonymous mutations within driver genes at low mutational burdens.

**Figure supplement 5.** Distribution of mutation rates of simulated tumors.

**Figure supplement 6.** Relative contribution of genetic hitchhiking and Muller's ratchet to fix deleterious passengers.

(*Figure 3D*). We tested whether positive selection on synonymous mutations within driver genes could explain this discrepancy. Indeed, we find that a model incorporating synonymous drivers agrees modestly better with observed statistics (3.5-fold relative likelihood, ABC posterior probability). The best-fitting model predicts that ~6% of synonymous mutations within driver genes experience positive selection, which is consistent with previous estimates for human oncogenes (*Supek et al., 2014*) (Materials and methods, *Figure 3D* and *Figure 3—figure supplement 4*). Furthermore, we observe additional evidence of selection and codon bias in synonymous drivers exclusive to low mutational burdens (TCGA samples, Materials and methods, *Figure 3—figure supplement 4*).

We note that although deleterious passengers are necessary to explain attenuation of negative selection with mutational burden in passengers, alternative explanations could also contribute to attenuation of positive selection in drivers. Specifically, high mutational burden tumors are more likely to contain mutations in pan-cancer driver gene sets which might not directly contribute to tumorigenesis in specific tumors, and thus might not be under direct positive selection in all tumors. Similarly, additional driver mutations might not directly contribute to tumor fitness beyond a certain number of driver mutations (e.g. 5-hit model). Nonetheless, it's important to note that Hill-Robertson interference is capable of reproducing all the features of the data (steep attenuation of negative selection in passengers and gradual attenuation of positive selection in drivers).

Overall, our results indicate that rapid adaptation through natural selection – acting on entire genomes, rather than individual mutations – is pervasive in all tumors, including those with elevated mutational burdens. Given the quantity of drivers and passengers observed in a typical cancer (TCGA), our model implies that cancer cells are in total ~90% fitter than normal tissues (119% total benefit of drivers, 46% total cost of passengers). A median of five drivers each of which has a mean benefit of ~19% accumulate per tumor in these simulations – also consistent with estimates from age incidence curves (*National Cancer Institute, 2007*), known hallmarks of cancer (*Hanahan and Weinberg, 2000*), and estimates of the selective benefit of individual drivers (*Dai et al., 2007*). Lastly, the mutation rates of tumors that could progress to cancer in our model also recapitulate observed mutation rates in human cancer (*Camps et al., 2007*) (median $3.7 \times 10^{-9}$, 95% interval $1.1 \times 10^{-10}$–$8.2 \times 10^{-8}$, *Figure 3—figure supplement 5*).

Most notably, under our modeling assumptions, all passengers together confer a fitness cost of ~46% per tumor. While this collective burden appears large, the individual fitness effects of accumulated passengers in these simulations (mean 0.8%) are similar to observed fitness costs in cancer cell lines (1–3%) (*Williams et al., 2008*) and the human germline (0.5%) (*Cassa et al., 2017*). Note that in our model, these passengers accumulated primarily via Muller's ratchet, while only ~5% accumulated via hitchhiking inferred using population genetics theory (*McFarland et al., 2013*) and MLE fitness effects, Materials and methods, *Figure 3—figure supplement 6*. These results suggest that Hill-Robertson interference is a plausible model for the empirical patterns of attenuated selection with mutational burden observed in the data.

## Discussion

Here, we argue that signals of selection are largely absent in cancer because of the inefficiency of selection and not because of weakened selective pressures. In low mutational burden tumors (≤3 total substitutions per tumor), increased selection for drivers and against passengers is observed and

ubiquitous: in SNVs and CNAs; in heterozygous, homozygous, clonal, and subclonal mutations; and in mutations predicted to be functionally consequential. These trends are not specific to essential or housekeeping genes. Importantly, these patterns persist across broad and specific tumor subtypes. Collectively, these results suggest that inefficient selection is generic to tumor evolution and that deleterious load is a nearly universal hallmark of cancer.

Importantly, these patterns of selection are missed when *dN/dS* rates are not stratified by mutational burden. Since <0.1% of mutations in TCGA reside within low mutational burden tumors (~1% of all tumors, *N*=83), *dN/dS* in passengers at low mutational burdens (~0.56) does not appreciably alter pan-cancer *dN/dS* of passengers (0.97 in our study, 0.82–0.98 in *Martincorena et al., 2017*; *Weghorn and Sunyaev, 2017*; *Zapata et al., 2018*; *Ostrow et al., 2014*). In fact, the power to detect negative selection on passengers at low mutational burdens is only possible by aggregating all mutations within these tumors and estimating *dN/dS* jointly. Thus, we believe that low mutational burden tumors are uniquely valuable for identifying genes and pathways under positive and negative selection. While only ~1% of tumors exhibit substantial negative selection, selection in drivers, selection on CNAs, and expression patterns of chaperones and proteasome components all show a continuous response to deleterious passenger load across a broad range of mutational burdens. Collectively, this suggests that passengers continue to be deleterious even in high mutational burden tumors.

Using a simple evolutionary model, we show that Hill-Robertson interference alone can explain this ubiquitous trend of attenuated selection in both drivers and passengers. *dN/dS* rates attenuate in drivers because the background fitness of a clone becomes more important than the fitness effects of an additional driver at elevated mutation rates. Furthermore, these simulations indicate that, despite *dN/dS* patterns approaching 1 in tumors with elevated mutational burdens, passengers are not effectively neutral (*Ns* > 1). Instead, passengers confer an individually weak, but collectively substantial fitness cost of ~46% that measurably impacts tumor progression. Because this simple evolutionary model does not explicitly incorporate many known aspects of tumor biology (e.g. haploinsufficiency, see *Supplementary file 1*), these fitness estimates are highly provisional. Nonetheless, we note that selection's efficiency in cancer is further reduced when spatial constraints are considered (*Sottoriva et al., 2015*).

The functional explanation for why passengers in cancer are deleterious is unknown. In germline evolution, mutations are believed to be primarily deleterious because of protein misfolding (*Drummond and Wilke, 2008*; *Lobkovsky et al., 2010*). Deleterious passengers in somatic cells should confer similar effects. Indeed, we find that elevated mutational burden tumors may buffer the cost of deleterious mutations by upregulating multiple heat shock pathways. However, deleterious passengers may carry other costs to cancers or be buffered by additional mechanisms. Understanding and identifying how tumors manage this deleterious burden should identify new cancer vulnerabilities that enable new therapies and better target existing ones (*Gorgoulis et al., 2018*; *Dai et al., 2007*; *Glaire and Church, 2017*).

## Materials and methods
### Defining mutational burden in SNVs and binning tumors

Since TCGA is composed of whole-exome data, which limits our ability to accurately assess mutations in non-coding regions, we elected to use the total number of protein-coding mutations (i.e. missense, nonsense, and synonymous mutations) as our proxy for the mutational burden of tumors. This allows us to focus on the highest quality set of mutations that we have, which can impact the power to detect selection (*Figure 2—figure supplement 15*). We note that this high-quality set mutations does not have evidence of germline contamination by common SNPs (MAF > 5%) from 1000 Genomes Project (*1000 Genomes Project Consortium et al., 2015*) (v2015 Aug) using ANNOVAR (*Wang et al., 2010*) to annotate mutations in TCGA (*Figure 2—figure supplement 4*). For all analyses calculating *dN/dS* in tumors stratified by their mutational burden, all variants within each bin of tumors were pooled together and *dN/dS* was calculated jointly on each bin of tumors. Counts of the number of mutations use to estimate *dN/dS* in each mutational burden can be found in *Figure 2—figure supplement 16*.

## A non-parametric null model of mutagenesis to calculate *dN/dS*

We assume that for any particular tumor, mutation rates are constant across a gene for a particular tri-nucleotide context and base change (e.g. C>G). Our procedure is inspired by constrained marginal models (or 'edge switching' in network analysis), whereby the marginal distributions of observations aggregated over known confounding variables are preserved under permutation to create a null distribution. In our application of this strategy, the marginal distributions of mutations (across tri-nucleotide context, base change, gene, and tumor) remain preserved – as they would be in a constrained marginal model; however, we exhaustively consider every acceptable permutation of the data. Because our approach is highly constrained, these permutations are exhaustively computable (median 36 alternatives per mutation). Thus, resampling is unnecessary.

Our null model presumes that all mutations of type *i*, defined by a tri-nucleotide context and base change, arise with probability $M_{igt}$ within each gene *g* and tumor *t*. For each gene, we tally the total quantity of nonsynonymous mutations $N_{ig}$ and synonymous mutations $S_{ig}$. Suppose selection enriches or depletes nonsynonymous mutations within a gene and tumor by a rate $\omega_{gt}$. The expected number of nonsynonymous and synonymous mutations within a particular tumor and gene are $E\left[d_N\right] = \omega \sum_i M_{igt} N_{ig}$ and $E\left[d_S\right] = \sum_i M_{igt} S_{ig}$ in the absence of selective pressures on synonymous mutations. As with the main text, $d_N$ and $d_N^{(observed)}$ are used interchangeably. Although $M_{igt}$ is unknown, *dN/dS* statistics attempt to infer selection nonetheless by noting that:

$$\frac{E\left[d_N\right]}{E\left[d_S\right]} = \frac{\omega_{gt} \sum_i M_{igt} N_{ig}}{\sum_i M_{igt} S_{igt}} = \omega_{gt} \frac{<M_{igt}, N_{igt}>}{<M_{igt}, S_{igt}>} = \omega_{gt} \frac{\rho_{MN}\|M_{gt}\|\,\|N_{gt}\|}{\rho_{MS}\|M_{gt}\|\,\|S_{gt}\|} = \omega_{gt} \frac{\rho_{MN}\|N_{gt}\|}{\rho_{MS}\|S_{gt}\|}$$

Note that $\rho_{AB} = A, B > \big(\|A\|\,\|B\|\big)$, where $\|A\| = \sqrt{<A,A>}$ is the Pearson product-moment correlation coefficient. When $\rho_{MN} \approx \rho_{MS}$,

$$\frac{E\left[d_N\right]/\|N\|_i}{E\left[d_S\right]/\|S\|_i} \approx \omega$$

That is, *dN/dS* is approximately equal to the selective pressures on nonsynonymous mutations when the accessible nonsynonymous and synonymous loci are properly accounted and when the correlation between mutational processes and nonsynonymous loci are roughly equivalent to the correlation between mutational processes and synonymous loci. Traditionally, this assumption was used to calculate *dN/dS*. To improve resolution of *dN/dS*, researchers have attempted to account for these correlations using sophisticated parametric models of $M_{igt}$. An alternative statistical approach, however, is to treat these correlations as nuisance parameters.

Constrained marginal models permute observed data in all possible manners that preserve the underlying covariance structure of the data (e.g. $\rho_{MN}$, $\rho_{MS}$). In our particular case of this method, we note that by definition, $d_N^{permuted} = \sum_i (d_N^{observed}{}_i N_i + d_S^{observed}{}_i N_i)$. Thus:

$$\frac{E\left[d_N^{permuted}\right]}{E\left[d_S^{permuted}\right]} = \frac{\sum_i \left(\omega M_{igt} N_{igt}^2 + M_{igt} N_{igt} S_{igt}\right)}{\sum_i \left(\omega M_{igt} N_{igt} S_{igt} + M_{igt} S_{igt}^2\right)}$$
$$= \frac{\omega \rho_{MN}\|M\|\,\|N\|^2 + \rho_{MN}\|M\|\,\|N\|\,\|S\|}{\omega \rho_{MS}\|M\|\,\|S\|\,\|N\| + \rho_{MS}\|M\|\,\|S\|^2}$$
$$= \frac{\rho_{MN}\|N\|}{\rho_{MS}\|S\|}$$

Hence, by dividing the observed mutations by all permutations, we eliminate the covariance of mutational processes with available loci and, thus, measure $\omega_{gt}$ directly for any particular gene-tumor combination without mutational bias.

Unfortunately, because of the log-sum inequality, mutational bias can arise once cohorts of genes and cohorts of tumor samples are binned. This problem is common to all *dN/dS* measures and is a consequence of the correlation of mutational biases with *selection* (i.e. $M_{igt}, \omega >$) – not the correlation of mutational biases with one another, as these covariances are already accounted for in a constrained marginal model. For example, if tri-nucleotide biases covary linearly with gene-level biases, and are independent of tumor-level biases, then a parametric estimate of $M_{igt}$ may deconstruct $M_{igt}$ into $M_{igt} = f\left(i, g, t, \rho_{ig}\right)$, where $\rho_{ig}$ is the covariation of tri-nucleotide mutational biases with gene-level biases. Nonetheless, $M_{igt}, \omega > \propto < \rho_{ig}, \omega >$ will still be ignored. Indeed, this covariation of mutational processes with selective forces is the focus of our current study: selection and genome-wide mutation

rate are correlated (i.e. $\sum_t M_{igt}\omega \neq 0$) because of Hill-Robertson interference. Hence, the level at which observed $d_N$ values $d_S$ are binned necessarily ignores covariation between mutational processes and selection (in addition to any variation of $\omega_{gt}$ within the bin). Another example of this binning challenge arises when positive and negative selection act on different regions of the same gene, which gene-level *dN/dS* binning can misinterpret as neutral evolution.

## Validation of non-parametric null model

To confirm that our null model can accurately estimate *dN/dS* even in the presence of extreme tri-nucleotide mutational biases, we simulated artificial data where different COSMIC signatures (*Tate et al., 2019Forbes et al., 2008*; *Forbes et al., 2008*) (SBS Signatures 1–9, v3) contribute to all of the mutations. Permuted $d_N$ and $d_S$ tallies for each mutational context were simulated by randomly sampling 1000 genes with the same mutational context. The fraction of permuted $d_N$ and $d_S$ tallies for each mutational context was used as weighted probabilities to derive observed $d_N$ and $d_S$ tallies. To simulate negative selection, $d_N$ counts were randomly removed from each context at a rate $1 - \omega_{gt}$ (e.g. a simulated 'true' *dN/dS* of 0.8 in a cohort of samples indicates a 20% chance of nonsynonymous mutations being removed in the samples). These simulated (true) rates were then compared to observed and permuted $d_N$ and $d_S$ tallies according to the *dN/dS* metric that we used throughout this study:

$$\frac{dN}{dS} = \frac{d_N^{(observed)}/d_N^{(permuted)}}{d_S^{(observed)}/d_S^{(permuted)}}$$

We confirmed that this approach accurately measures selection in the presence of simulated mutational biases (*Figure 2—figure supplement 2*).

Lastly, we note that binning nonsynonymous and synonymous mutations at the genome-wide level (e.g. drivers and passengers) provided the most robust estimates of *dN/dS* when bootstrapping observed tumor samples. Statistical power is insufficient when binning at the individual gene level. Bootstrapping also demonstrated that log-transformation of *dN/dS* values increases statistical power, and thus was generally applied to *dN/dS* analyses in this study.

## A parametric null model of mutagenesis

For comparison, we also calculated *dN/dS* using *dNdScv* (*Campbell and Martincorena, 2017*) – a previously published parametric null model of mutagenesis in cancer (*Martincorena et al., 2017*). To compare both methods, *dNdScv* ran globally and separately on samples stratified by the total number of substitutions using the following parameters: max_coding_muts_per_sample = Inf max_muts_per_gene_per_sample = Inf.

Global *dN/dS* values of all nonsynonymous mutations ($w_{all}$, reported by *dNdScv*) were used. This model reproduced our non-parametric *dN/dS* trends (*Figure 2A*) and was used to infer patterns of selection in synonymous mutations (*Figure 3—figure supplement 4*). We note that stratifying tumors in TCGA into 20 bins of equal sample size (as was done in *Martincorena et al., 2017*), rather than evenly spaced bins, averages out a significant proportion of the negative selection observed in passengers, since low mutation burden tumors reside within the tail-end of the distribution (*Figure 2—figure supplement 7*).

## Identification of driver genes in cancer

For all analysis using SNVs, unless explicitly stated, a comprehensive list of 299 pan-cancer driver genes derived from 26 computational tools was used to catalog driver genes (*Bailey et al., 2018*). Other pan-cancer driver gene sets tested were derived from COSMIC's Driver Gene Census (*Tate et al., 2019*; *Forbes et al., 2008*) (downloaded on October 2016) and IntOGen's Cancer Drivers Database (*Gonzalez-Perez et al., 2013*) (v2014.12) which contained 602 and 459 number of driver genes, respectively.

Many driver genes are associated with only particular tumor subtypes. To compare patterns of selection across cancer subtypes without increasing or decreasing the size of the list for each subtype, we chose to use a single set of driver genes for most analyses. This may understate the degree of positive selection in driver genes as mutations in these genes may be passengers in some tumor subtypes. In *Figure 2—figure supplement 5*, we investigate patterns of selection using the top 100 driver

genes identified for each tumor type and observe decreased signatures of positive selection overall in driver genes. Nevertheless, the patterns of attenuated selection in drivers and passengers remain. While tissue-type specific driver genes certainly exist, our results suggest that our statistical power to detect drivers still remains too limited to justify subdividing analyses by tumor type in many cases.

For all CNA analysis, GISTIC 2.0 *Mermel et al., 2011* was used to identify a set of genomic regions enriched for copy number gains and copy number losses using recommended settings with a confidence threshold of 0.9. CNAs used to identify these peaks were downloaded from the NIH Genomic Data Commons (GDC) (*Grossman et al., 2016*) in the TCGA cohort. For each amplification peak, the closest gene was annotated as a putative oncogene, and similarly the closest gene to each deletion peak was annotated as a putative tumor suppressor. The top 100 amplification peaks (oncogenes) and deletion peaks (tumor suppressors) were classified as drivers for each of the 32 tumor types. Thirty-four percent of identified driver genes appear in more than one tumor type, while 2.6% of identified driver genes appear in more than five tumor types.

For both SNV and CNA analysis, passengers were defined as mutations that did not reside within driver genes. The vast majority of mutations are passengers, and their relative totals for both SNVs and CNAs are depicted in *Figure 2—figure supplement 16*.

## Annotation of clonal and subclonal mutations

VAFs were calculated per site as the number of mutant read counts divided by the total number of read counts. VAFs were adjusted for purity using calls made by ABSOLUTE (*Grossman et al., 2016*; *Carter et al., 2012*), collected from GDC. A VAF threshold of 0.2 was used to define 'subclonal' (<0.2) vs. 'clonal' (>0.2) SNVs. Different VAF thresholds were considered (*Figure 2—figure supplement 10*) and the choice of 'clonal' thresholding did not impact the conclusions of this study.

## PolyPhen2 analysis

PolyPhen2 annotations in the MC3 SNP calls were used (*Adzhubei et al., 2010*). Only missense mutations that were categorized as either 'benign', 'probably damaging', or 'possibly damaging' were used. The fraction of pathogenic missense mutations was calculated as the number of pathogenic mutations categorized as either 'probably damaging' or 'possibly damaging' divided by the total number of categorized mutations.

## Classification of genes by functional category

To test for patterns of selection in functionally related genes, we annotated all mutations by different functional categories and GO terms (*Harris et al., 2004*). Oncogenes and tumor suppressors were annotated from a curated set of 99 high confidence cancer genes (*Kumar et al., 2015*). Essential genes were collected from a genome-wide CRISPR screen that identified genes required for proliferation and survival in a human cancer cell line (*Wang et al., 2015*). Housekeeping genes were defined as genes with an exon that is expressed in all tissues at any non-zero level, and exhibits a uniform expression level across tissues (*Eisenberg and Levanon, 2015*). Interacting proteins were downloaded from the mentha database in April 2019 (*Calderone and Cesareni, 2012*).

To identify highly expressed genes, median transcripts per million (TPM) in 54 tissue types (v7 release) were downloaded from the Genotype-Tissue Expression (GTEx) project (*GTEx Consortium, 2020*; *Carithers and Moore, 2015*). Tissues that contained high expression in most genes, specifically testes, were removed. Only genes that had TPM counts above zero in any of the 53 remaining tissues were used. TPM counts were averaged across all tissues. Highly expressed genes were defined as the top 1000 genes expressed across all tissues.

To test for signals of negative selection in other functional groups, we annotated mutations by candidate GO terms according to biological processes: Transcription Regulation (GO Term ID: 0140110), Translation Regulation (GO Term ID: 0045182), and Chromosome Segregation (GO Term ID: 0007059).

## Somatic CNAs

All CNAs were downloaded from the COSMIC database on June 2015 (*Tate et al., 2019*; *Forbes et al., 2008*). Mitochondrial CNAs were discarded from analysis, as copy number changes are difficult to infer. Gene annotations and the locations of telomeres and centromeres were downloaded

from the UCSC Genome Browser (hg19). Telomeric and centromeric regions were masked from all measurements of $dE/dI$. Because the selection patterns of non-focal CNAs – alterations with at least one terminus in a telomere or centromeric region – were not noticeably different from long (>100 kb) focal CNAs, these two alteration classes were aggregated for analysis. Notably, we observed positive selection for both amplifications and deletions within oncogenes, and for both deletions and amplifications within tumor suppressors. For this reason, we did not distinguish between gains and losses, nor oncogenes and tumor suppressors in published analyses: any CNA that overlapped an oncogene or tumor suppressor in any region (for any fraction of the CNA) was classified as a driver. Mutational burden was defined simply as the total number of CNAs within a sample. Pan-cancer CNAs from cBioPortal (August 2018) were also analyzed, however consistent purity and ploidy estimates could not be obtained by using either ABSOLUTE (*Carter et al., 2012*) or TITAN (*Ha et al., 2015*), so this data was not used for published analyses of CNAs.

## Measurements of selection on CNAs

$dE/dI$ was calculated using a 'breakpoint frequency' metric and a 'fractional overlap' metric. For both metrics, the $dE/dI$ of a particular gene set $i$ (e.g. driver or passenger genes) is defined by a genomic track $T_{i,g}$, which is one for every annotated region $g$ of the track and zero elsewhere. Only non-centromeric and non-telomeric regions are considered in the mappable human genome $G$. Each CNA $C_{g,m}$ is defined by its position on the genome $g$ and the mutational burden $m$ of the tumor harboring the mutation. For 'breakpoint frequency' $C_{m,i}$ is one at the position of both termini of the CNA and zero elsewhere. For 'fractional overlap' $C_{m,i}$ is $1/L$, where $L$ is the length of the CNA, for every region of the genome spanned by the CNA and zero elsewhere. For a particular range of mutational burdens $M$, $dE/dI$ was defined as:

$$\frac{dE}{dI}_{i,M} = \frac{\sum_m^M \sum_g^G T_{i,g} C_{m,g}}{\sum_g^G T_{i,g}}$$

We note that calculation is accelerated by >×100 by commuting $T_{i,g}$ with the outer summation ($\Sigma_m^M$). Lastly, we randomly permuted the start and stop positions of each CNA, while preserving its length, to derive a set of neutral CNAs not experiencing selection. This permutation analysis finds that $dE/dI$ for both breakpoint frequency and fractional overlap is ~1 in the absence of selection (*Figure 2—figure supplement 8*).

## Tumor purity analysis in TCGA samples

Tumor purity estimates from the ABSOLUTE algorithm (*Carter et al., 2012*) were downloaded from the GDC on May 2020. To evaluate the effects of tumor purity on patterns of selection, tumors below increasing thresholds of tumor purity were removed from the analysis, and $dN/dS$ was calculated on tumors stratified by mutational burden bins (as described above.)

## Expression analysis

Gene expression data was downloaded from the COSMIC database on September 2019. Genes used to identify different protein folding pathways were downloaded from *Kampinga et al., 2009*, genes involved in protein degradation pathways were identified from *Tanaka, 2009*. The median gene expression of all genes in each protein folding pathway was used. Patients were binned by the total number of substitutions (using MC3 SNP calls from TCGA) and CNAs, and the average gene expression of each bin was calculated.

## Cancer subtype analysis

All tumor subtypes in were grouped into nine sub-categories, based on broad, predominantly anatomical features. Anatomical features (i.e. organ and systems of organs), rather than histological features or inferred cell-of-origin, were used as groupings because we believe that the fitness effects of mutations should be predominantly defined by the environment of the tumor. Nevertheless, we observed attenuated selection in both drivers and passengers in many broad histologically defined classifications (e.g. adenocarcinomas and sarcomas). For all cancer grouping analysis (broad and subtype), tumors were stratified into bins by the total number of substitutions ($d_N+d_S$) on a log-scale. Since tumor subtypes vary in their range of mutational burdens, (e.g. KIRC cancer subtypes only have

tumors with <100 substitutions), *dN/dS* values in the lowest and highest mutational burden bin for each cancer subtype are shown.

Specific cancer subtype categories were taken directly from the NCI GDC (*Grossman et al., 2016*). Because CNAs were downloaded from COSMIC, CNA datasets were not classified with this same ontology. *Supplementary file 2* details how CNA classifications were mapped on GDC categories (and sometimes more broadly defined groups). All subtypes with >200 samples were used in our CNA subtype analyses (*Figure 2—figure supplement 13*).

## An evolutionary model with Hill-Robertson interference

Somatic cells in our populations are modeled as individual cells that can stochastically divide and die in a first-order (memoryless) Gillespie algorithm. This model was developed and described previously (*McFarland et al., 2014*). During division, cells can acquire advantageous drivers with rate $\mu T_{drivers}$ and deleterious passengers with rate $\mu T_{passengers}$ – these values specify the mean of Poisson-distributed pseudo-random number (PRN) generators that prescribe the number of drivers and passengers conferred during division (e.g. the number of drivers per division $n_d = \text{Poisson}[n_d = k; \lambda = \mu T_{drivers}] = \lambda^k e^{-k}/k!$). The DFEs conferred by each driver and each passenger are exponentially distributed PRNs with probability densities $P(s_i = x; s_{drivers}) = \text{Exp}[-x/s_{drivers}]/s_{drivers}$ and $P(s_i = x; s_{passengers}) = -\text{Exp}[-x/s_{passengers}]/s_{passengers}$, respectively. Simulations with other exponential-family DFEs do not qualitatively differ from these exponential distributions (*McFarland et al., 2013*). The aggregate absolute cellular fitness is $f = \prod_i^{\text{all mutations}} (1 + s_i)$ in our multiplicative epistasis model and $\Delta f = s_i/(1 + \nu f)$ with $\nu = 1$ in our diminishing-returns epistasis model, where $\Delta f$ is the change in cellular fitness with each mutation (*Arjan et al., 1999*). The rate of cell birth is inversely proportional to cellular fitness, while the rate of cell death $D(N; N^0) = \text{Log}\left[1 + \frac{N}{(e-1)N^0}\right]$ increases with the population size of the tumor $N$. With these birth and death processes, mean population size abides by a Gompertzian growth law in the absence of additional mutations, which is scaled by the mean cellular fitness $\text{E}[N(<f>)] = \log[1 + <f>/N^0]$ (derived from master equation *McFarland et al., 2013*). While, programmatically, mutations exclusively affect the birth rate and the constraints on growth exclusively affect the death rate, we previously demonstrated that birth and death rates are generally nearly balanced such that dynamics are not affected by this design choice.

Because somatic cells do not recombine during cell division, dominance coefficients were not explicitly modeled. Thus in diploid cancers, our selection coefficients estimate the mean heterozygous effect of drivers and passenger (i.e. *hs*). Similarly, loss of heterozygosity (LOH) events (gene losses, gene conversions, mitotic recombination, etc.) are not explicitly modeled either; however, these events can be viewed as additional mutations that may be either adaptive drivers or deleterious passengers in the model. As sequencing data improves, we believe that it will be informative to explicitly model dominance coefficients, tumor ploidy, and LOH events.

Simulations progressed until tumor extinction ($N=0$ cells), malignant transformation ($N=10^6$ cells), or until ~100 years had passed (18,500 generations). Only fixed mutations (present in the most recent common ancestor) within clinically detectable growths were analyzed in our ABC pipeline. The behavior of this model has been described previously (*McFarland et al., 2013*; *McFarland et al., 2014*) and the most relevant assumptions of this model and their effects on the conclusions of this study are described in *Supplementary file 1*.

Cells in our populations are fully described by their accrued mutations, and birth and death times. Birth and death events were modeled using an implementation of the next reaction (*Gibson and Bruck, 2000*), a Gillespie algorithm that orders events using a heap queue. Generation time in our model was defined as the inverse of the mean birth rate of the population: $1/<B(d, p)>$. While all mutation events occurred during cell division, if mutations were to occur per unit of time (rather than per generation), rapidly growing tumors would acquire drivers at a slightly slower rate as generation times decline over time. This effect, however, is negligible compared to the variation in waiting times conferred by the variation in mutation rates (division times merely double, while mutation rates vary by 100,000-fold).

This simple evolutionary model is defined by five parameters $\mu T_{drivers}$, $\mu T_{passengers}$, $s_{drivers}$, $s_{passengers}$, and $N^0$. The target size of drivers is defined as the approximate number of nonsynonymous mutations in the Bailey driver screen $T_{drivers}$ = (# of driver genes)·(mean driver length)·(fraction of SNVs that are nonsynonymous)=300 genes · 1298 loci/gene · 0.737 nonsynonymous loci/ loci = 286,886

nonsynonymous loci. The target size of passengers was simply the remaining loci in the protein coding genome, $T_{passengers}$ = 20,451,136 nonsynonymous loci. The mutation rate was constant throughout each tumor simulation and randomly sampled from a uniform distribution in log-space that ranged from $10^{-12}$ to $10^{-7}$ mutations·loci$^{-1}$·generation$^{-1}$. While tumors were initiated from this broad range, malignancies ($N > 10^6$ cells) were almost always restricted to mutation rates between $10^{-10}$ and $10^{-8}$ (*Figure 3—figure supplement 5*), as tumors with mutation rates drawn below this range almost never progressed to cancer within 100 years and tumors with mutation rates drawn above this range went extinct through natural selection.

The likelihood that tumors progress to cancer in the presence of deleterious passengers depends heavily on the initial population size $N^0$ of the tumor. This dependence was studied previously (*McFarland et al., 2014*), where it was demonstrated that reasonable evolutionary simulations (those that progress to cancer >10% of the time, but <90% of the time) are restricted to a four-dimensional manifold $N^*$ within the five-dimensional phase space of parameters. For this reason, $N^0 = N^*(s_{drivers}, s_{passengers}, \mu T_{drivers}, \mu T_{passengers})$ was determined by the other four parameters. To first order, this manifold is $T_{passengers} s_{passengers} / (T_{drivers} s_{drivers})$ (*Weghorn and Sunyaev, 2017*), however a more precise estimate (Equation S8 of *McFarland et al., 2014*) incorporating more precise estimates of Muller's ratchet and the effects of hitchhiking on both driver and passenger accumulation rates, which does not exist in closed form was used. Additionally, at very low values of $s_{drivers}$, progression to cancer is limited by time, not by the accumulation of deleterious passengers. Hence, we assigned $N^0$ such that:

$$N^0 = Max_{N^0[P_{cancer}(N^0/N^*)=0.5, \overline{t_{cancer}}(N^0/N^*)=18,500 \, generations]}$$

Here, $P_{cancer}$ and $t_{cancer}$ – the likelihood and waiting time to cancer – are defined by Equation S8 and S12 respectively in *McFarland et al., 2014*. $N^0$ was determined from these equations using Brent's method. *Figure 3—figure supplement 2* depicts the values of $N^0$, which ranged from 1 to 100 for all simulations.

In tumors that progress to malignancy ($N = 10^6$), only fixed nonsynonymous mutations (present in all simulated cells) were recorded. We also recorded (i) the fitness effect of these mutations, (ii) the mean population fitness, (iii) the number of generations until malignancy, and (iv) the mutation rate. These two values were used to generate the number of synonymous drivers and passengers, where $P(d_s = k) = Poisson[k; \lambda = \mu T_{drivers/passengers}/r \, t_{MRCA}]$ defines the number of synonymous drivers/passengers conferred, $t_{MRCA}$ represents the number of division until the most recent common ancestor arose in the simulation, $r = 2.795$ represents the ratio of nonsynonymous to synonymous loci within the genome, weighted by the genome-wide tri-nucleotide somatic mutation rate, and the Poisson PRN generator was defined above. In simulations where synonymous drivers could arise, a fraction of the recorded nonsynonymous mutations (ranging from 0% to 20%) were simply re-labeled as synonymous drivers (as opposed to nonsynonymous drivers). This was done, again, by Poisson sampling in proportion to the desired fraction for each cancer simulation.

20×20 combinations of $s_{drivers}$ and $s_{passengers}$ parameters were simulated (*Figure 3—figure supplements 1–2*). Simulations were repeated until 10,000 cancers at each parameter combination were obtained or until 10 million tumor populations were simulated. While we attempted to initiate tumors at a population size where the probability of progression to cancer was 50%, some parameter combinations still did not yield 10,000 cancers after 10 million attempts (i.e. $P_{cancer} < 0.1\%$). These combinations were predominately at low values of $s_{drivers}$, which were far from the MLE estimate of $s_{drivers}$ and represent unrealistic evolutionary scenarios: drivers cannot be weakly beneficial, relegated to only 300 genes, and still overcome deleterious passengers within 100 years. These simulations are annotated as 'progression impossible'. Simulation parameter sweeps were performed for both the multiplicative and diminishing returns epistasis models. Twenty fractions of synonymous drivers were also generated (ranging from 0% to 20%). These fractions were generated by simply re-labeling the driver mutations which conferred fitness (generated during the simulation) as synonymous, instead of nonsynonymous.

## Summary statistics of simulated and observed tumors

For both simulated and observed data, we summarized $dN/dS$ rates vs. mutational burden for drivers and for passengers by decade-sized bins: (0, 10], (10, 100], (100, 1,000]. Mutational burden for simulations was defined as the total number of substitutions ($d_N + d_S$) – exactly as it was defined for observed

data. For simulated data, $dN/dS = d_N/(d_S \cdot r)$. Like observed data, $dN/dS$ rates attenuated toward 1 for both drivers and passengers for all values of $s_{drivers}$ and $s_{passengers}$.

Mutational burdens (MB) for simulated and observed data were summarized with the parameters of a negative binomial distribution, where $P(\text{MB} = k; n, p) = \begin{pmatrix} k + n - 1 \\ n - 1 \end{pmatrix} p^n (1 - p)^k$. This distribution has been used previously to summarize the MB of human tumors (*Turajlic et al., 2012*) and exactly defines the expected number of mutations at transformation in a multi-stage model of tumorigenesis (*Frank, 2007*) when $n$ drivers are needed for transformation and the probability that any mutation be a driver is $1 - p$ (*Michor et al., 2005*). Both $n$ and $p$ were used to summarize MB. These quantities were determined by maximum likelihood optimization of the probability mass function above over the support of mutational burdens of [1, 1,000] substitutions. The Han-Powell quasi-Newton least-squares method was used for optimization.

Age-dependent cancer incidence rates (CI) were summarized with the parameters of a gamma distribution, where $P(CI \leq t; k, \theta) = \frac{1}{\Gamma(k)} \gamma\left(k, \frac{t}{\theta}\right)$. Here, $\gamma(s, x) = \int_0^x t^{s-1} e^{-t} dt$ is the lower incomplete gamma function and $\Gamma(k) = \gamma(k, \infty)$ is the regular gamma function. Similar to our summarization of mutational burdens, this distribution is a generalization of the exact waiting time to transformation expected from a multi-stage model of tumorigenesis when tumors arise at a uniform rate over time, require $k$ drivers for transformation, and wait an average time of $\theta$ between drivers (*Michor et al., 2005*). This cumulative distribution function was fit to observed incidence rates for all patients above 20 years of age using the least squares numerical optimization defined above (all cancer sites combined, both sexes, all races, 2012–2016; *Howlader, 2013*). Patients under 20 years of age were excluded because cancers in these patients generally arise from germline predispositions to cancer, which are (i) not directly modeled by our simulations, (ii) not detected as somatic mutations, and (iii) result in age incidence curves that do not agree with a gamma distribution (*Frank, 2007*). Because all cancer simulations are initiated at $t=0$ (instead of uniformly in time, as is presumed in the multi-stage model), the simulated data was fit using the probability density function of this distribution (instantaneous derivative) using maximum likelihood and the optimization algorithm described above. The cumulative distribution, then, represents the expected age incidence cancer incidence rate when simulations begin at uniformly distributed moments in time and, thus, was used to generate *Figure 3D*. Only the shape parameter $k$ was used in ABC (and $\theta$ was ignored), as this parameter only specifies the dimensionality of time (simulation time was measured in cellular generations, not years) and all values of $\theta$ in our simulations are equivalent under a gauge transformation. Additionally, we do not expect the exact times of incidence to be particularly informative as the time of transformation is generally somewhat earlier than the time of detection.

## Use of ABC for model selection and parameter inference

Like many Bayesian analyses, the main steps of an ABC analysis scheme are: (i) formulating a model, (ii) fitting the model to data (parameter estimation), and (iii) improving the model by checking its fit (posterior-predictive checks), and (iv) comparing this model to other models (*Csilléry et al., 2012*; *Gelman, 2004*).

The nine summary statistics described above were used to compare simulations to observed data. Agreement was summarized with a log-Euclidean distance, as all summary statistics resided on the domain [0, ∞) and log-transformation of the summary statistics minimized heteroscedasticity of the simulated data relative to a square-root or no transformation. Variance of the summary statistics was not normalized. ABC was performed using the 'abc' R package (*Csilléry et al., 2012*).

The rejection method (feedforward neural net) and tolerance (0.5) were chosen based on their capacity to minimize prediction error of the simulated data using leave-one-out CV (*Figure 3—figure supplement 3*). Ten-thousand instances of the neural network, which was restricted to a single layer, were initiated and the median prediction of these networks were used. These parameters were used for both model comparison and parameter inference. For parameter inferencing, the $s_{drivers}$ and $s_{passengers}$ prior values were log-transformed.

For the synonymous driver model, the base model (without synonymous drivers) was simply the lowest quantity of synonymous drivers (0%) in the parameter sweep of synonymous driver quantities (*Figure 3—figure supplement 3B*). The posterior probability mass of this value 0.017 was used as the one-sided p-value for the null hypothesis that these two models are equally predictive. Although the

synonymous driver model agreed with the observed data slightly better, $s_{drivers}$ and $s_{passengers}$ parameters could not be inferred from the data because the potential for synonymous drivers destroys the utility of a $dN/dS$ statistics, which is predicated on the notion that synonymous mutations are neutral. Virtually any value of $dN/dS$ is attainable when the right combinations of selective pressures on nonsynonymous and synonymous are paired (*Figure 3—figure supplement 3C*).

## Code availability

All code for empirical analysis and generation of summary statistics are publicly available under the open-source MIT License at https://github.com/petrov-lab/cancer-HRI. (*Tilk, 2022a* copy archived at swh:1:rev:5d67f4a946e2d80efdc71c2ef689266678d8ff75). Code for simulations of tumor growth with advantageous drivers and deleterious passengers is also available at https://github.com/mirnylab/pdSim, (*Tilk, 2022b* copy archived at swh:1:rev:f08bd75aabf7213e253baf26d219c374a745c8d4).

## Acknowledgements

We thank Judith Frydman for her contribution on the heat shock response analysis, Monte Winslow for his contribution on cancer subtype analysis, Donate Weghorn for her contribution on the interdependence of $dN/dS$ and mutational burden, Leonid Mirny, Grant Kinsler, Gabor Boross, Chuan Li, Alison Feder, Eliot Cowan, and other members of the Petrov and Curtis labs for helpful comments and discussions. This work is supported by NIH grants T32-HG000044-21, E25-CA180993; the Director's Pioneer Award DP1-CA238296 to CC; R01-CA207133, R35-GM118165, and R01-CA231253 to DAP; and K99-CA226506 to CDM.

## Additional information

### Funding

| Funder | Grant reference number | Author |
|---|---|---|
| National Human Genome Research Institute | T32-HG000044-21 | Christopher D McFarland |
| National Institutes of Health | E25-CA180993 | Christopher D McFarland |
| National Institutes of Health | DP1-CA238296 | Christina Curtis |
| National Cancer Institute | R01-CA207133 | Dmitri A Petrov |
| National Institute of General Medical Sciences | R35-GM118165 | Dmitri A Petrov |
| National Institutes of Health | R01-CA231253 | Dmitri A Petrov |
| National Cancer Institute | K99-CA226506 | Christopher D McFarland |

The funders had no role in study design, data collection and interpretation, or the decision to submit the work for publication.

### Author contributions

Susanne Tilk, Conceptualization, Data curation, Software, Formal analysis, Funding acquisition, Validation, Investigation, Visualization, Methodology, Writing - original draft, Writing - review and editing; Svyatoslav Tkachenko, Validation; Christina Curtis, Supervision, Funding acquisition, Writing - review and editing; Dmitri A Petrov, Conceptualization, Supervision, Funding acquisition, Validation, Methodology, Writing - original draft, Writing - review and editing; Christopher D McFarland, Conceptualization, Software, Formal analysis, Supervision, Investigation, Visualization, Methodology, Writing - original draft, Writing - review and editing

### Author ORCIDs

Susanne Tilk (iD) http://orcid.org/0000-0002-9156-9360

Christina Curtis http://orcid.org/0000-0003-0166-3802
Dmitri A Petrov http://orcid.org/0000-0002-3664-9130

**Decision letter and Author response**
Decision letter https://doi.org/10.7554/eLife.67790.sa1
Author response https://doi.org/10.7554/eLife.67790.sa2

## Additional files

### Supplementary files
- MDAR checklist
- Supplementary file 1. Model assumptions of tumor evolution and its anticipated effects.
- Supplementary file 2. Broad (meta-categories) of cancer groupings used in Figure 2 and Figure 2—figure supplement 12-13.

### Data availability

Exonic, open-access SNV calls (WES) of 10,486 cancer patients in (The Cancer Genome Atlas) TCGA were downloaded from the Multi-Center Mutation Calling in Multiple Cancers (MC3) project. This repository uses a consensus of seven mutation-calling algorithms. Expression data of SNVs were downloaded from the Genotype-Tissue Expression (GTEx) project (v7 release). All CNAs were downloaded from the COSMIC database on June 2015.Gene expression data compared to CNAs was downloaded from the COSMIC database on 14 September 2019.

The following previously published datasets were used:

| Author(s) | Year | Dataset title | Dataset URL | Database and Identifier |
| --- | --- | --- | --- | --- |
| Ellrott K, Bailey MH, Saksena G | 2018 | TCGA - MC3 mutation calls | https://gdc.cancer.gov/about-data/publications/mc3-2017 | Genomic Data Commons, mc3-2017 |
| Tate JG | 2018 | COSMIC - gene expression data | https://cancer.sanger.ac.uk/cosmic/download | COSMIC, nar/gky1015 |
| GTEX Consortium | 2020 | GTEX - expression data | https://www.gtexportal.org/home/datasets | GTEX, phs000424.v8 |

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
