## [Editor Report]

This is an important paper that shows most cancers unavoidably accumulate damaging mutations. Whilst the majority of claims are convincingly supported by the data, evidence that damaging changes are buffered by heat shock pathways is currently incomplete. The insights into selection efficiency are important for the understanding of cancer growth and response to therapy. A broader implication is that high mutation load tumors may use common strategies to tolerate accumulated deleterious mutations, providing a therapeutic target.

---

## [Decision Letter]

**Decision letter after peer review:**

Thank you for submitting your article "Most cancers carry a substantial deleterious load due to Hill-Robertson interference" for consideration by *eLife*. Your article has been reviewed by 3 peer reviewers, including Martin Taylor as the Reviewing Editor and Reviewer #3, and the evaluation has been overseen by Molly Przeworski as the Senior Editor. The following individual involved in review of your submission has agreed to reveal their identity: Elena Kuzmin (Reviewer #1).

Essential revisions:

(1) The authors state that they are excluding tumors with either nonsynonymous(n)=0 or synonymous(s)=0 mutations. Since nonsynonymous and synonymous variants occur in a ratio of about 3:1, this exclusion of tumors would seem to lead to an inflation of the signal of selection in the first (lowest mutation) bins. Additional demonstration is required to show if this distorts the estimate of selection for low mutation burden tumours. For example, by adding pseudo-counts of mutations or by aggregation over tumours in the same mutation load "bin" as was performed for some analyses.

(2) The decline of dN/dS on driver genes is the subject of the supplementary text and we note the efforts taken to disentangle Hill-Robertson interference effects from other possible explanations for dN/dS decay in drivers with increasing tumor mutation burden (TMB). Driver genes can be quite tissue-specific (and thus mus-identified for a tumor) and number of drivers per tumour estimated to span approximately 1 to 10 (Martincorena et al., 2017). Consequently, the fitting of a single set of model parameters and showing they do not match well the observed data (Supplementary note figure) is insufficient to exclude the misidentification of driver genes, or presence of nonsynonymous-neutral mutations in annotated driver genes, as an explanation for the decline in dN/dS with increased TMB. We think it important that a range of justifiable parameters are applied in this modelling, to test if the observed data is robustly outside reasonable parametrisation of the model.

(3) In many figures that show dN/dS as a function of n+s (starting with Figure 2A and extending to Figures S2, 3, 9, 10, 12, 22 and 25), there are no error bars indicated, as opposed to the statement in the figure caption. The error bars/shading should be shown. In Figure 2A, is the observed depletion in the second bin still significant?

*Reviewer #1 (Recommendations for the authors):*

1. Figure panels should be called out sequentially. For example, Figure 2G is called out before Figure 2D. This happens throughout the text, including main and supplementary figures, and should be corrected.

2. Figure 2G shows that mean gene expression of genes encoding chaperones and the proteasome increases with increasing mutational burden. What about protein abundance? Is this in agreement with gene expression?

3. Figure 2 mentions error bars in the figure legend, but no panel displays error bars. This is also true for Figure S13 and other figures. Authors should display the error bars to which they are referring to make their analysis more convincing.

4. Pg. 9 line 295 describes results of the analysis across genes belonging to different GO terms. However, Figure S13 only shows 3 categories: chromosome segregation, transcription and translation. How were these categories chosen? What about other categories? Such cherry picking doesn't convincingly support the conclusions that no specific GO functions are enriched. Also, translational regulation shows higher dN/dS in low mutation tumors suggesting that there is positive selection for passengers in this category. Authors should discuss in their manuscript why this is the case.

5. Figure S15 shows the attenuation in selection of CNAs across cancer subtypes and broad cancer groups. However, HNSC and kidney cancer appear to be the exceptions. Authors should provide an explanation for these observations in the main text.

6. Generally, copy number variations are considered to be > 50 bp. Is there a rationale as to why authors chose 100 kb to be their cut-off in Figure 2C? If the size of CNA is an important parameter, then authors should explain why that is.

7. Non-allelic recombination and non-homologous recombination mechanisms involving replication accidents that lead to chromosome breakage occur with some frequency in somatic cells. How does the frequency of these events impact the selection efficiency in cancer as it relates to drivers and passengers? Can this also be incorporated in their evolutionary model?

8. Authors mentioned that haploinsufficiency was not used in the model. What about loss of heterozygosity which is extensive in cancer genomes? Can this parameter be included in the evolutionary model and how would it impact the results?

*Reviewer #2 (Recommendations for the authors):*

1. The authors have taken great care to study single-nucleotide variants and large CNAs. It would be great if they could confirm their findings by also showing the effect on small insertions and deletions.

2. Figure S5 is showing a bias in the determination of dN/dS from simulation results and the correlation between mutation rate and n+s. I am not sure I understand why dN/dS under a neutral simulation would be biased. Also, the low median correlation between n+s and the mutation rate (<0.4) is quite surprising. I would have expected these to be almost perfectly correlated. Likewise, I do not understand the formula after l. 631. It states that this is the joint density of the two Poisson random variables that denote nonsynonymous and synonymous mutation count, yet there is an additional unexplained factor in the denominator, which corresponds to the probability of s>0. If the simulation that underlies Figure S5 was also used in the ABC-based parameter inference, this would raise a serious cause for concern.

3. The simulation starts when the first mutation with positive selective effect initiates population growth, which can be very late in a patient's life. How does the assumption that up to 100 years can pass after this affect the parameter estimates?

4. To which extent does the inferred distribution of selection effects depend on the allowable parameter range? For example, s_passengers extends beyond the initially allowable range after the fit (Figure 3C).

5. It is not entirely clear to me how the partitioning of the likelihood between Muller's ratchet and hitchhiking vs other effects can be made and how robust these inferences are with respect to variation of the modeling assumptions (e.g. about initial population size or mode of selection). Is the necessity of inclusion of selected synonymous variants on driver genes a robust result or not, taking into account the discussion on p. 26f.?

6. In Figure S4, the authors report the correlation of n+s with other measures of tumor mutation load. Given the relative sizes of the different regions that are displayed, i.e. whole genome:intergenic:intronic:exonic:protein-coding, of roughly 100:60:40:2:1, the displayed numbers do not make sense, as their ratios are 100:100:100:1:0.001.

7. I am not sure I understood well how CNAs were analyzed. Based on the description in l. 669ff., it appears that putative cancer driver genes were identified from the CNA data based on recurrence. Were the same data then analyzed for CNAs falling into said putative cancer driver gene regions to infer selection? This would appear a bit circular.

8. I do not understand the formula shown after li. 738. It appears it is showing the fraction of genes that intersect a CNA boundary, summed over all tumors in a given n+s bin. Each CNA can be counted twice if both of its boundaries fall into a gene. Why is the mean value of this 1?

9. In all figures that show dN/dS as a function of n+s (starting with Figure 2A and extending to Figures S2, 3, 9, 10, 12, 22 and 25), there are no error bars indicated, as opposed to the statement in the figure caption. In Figure 2A, is the observed depletion in the second bin still significant?

10. In l. 290, I understand that the authors argue that differential dominance effects between heterozygous early- and late-arising mutations could be affecting the efficacy of selection on subclonal variants compared to clonal variants. I do not see this claim well motivated or corroborated.

11. In Figure 2D, the caption states that mutations have been separated into two groups by their clonality, yet the figure shows three curves. What do they correspond to? Are the results still significant given the partitioning of the mutation data into smaller subsets?

12. Figure 3 does not have a panel G.

*Reviewer #3 (Recommendations for the authors):*

I enjoyed reading the manuscript, it was well written, generally clear figures and very through provoking.

Only a small number of specific points to address:

Line 60.- The description of dN and dS here along with the interpretation of dN/dS=1 as neutral implies that you are just counting non-synonymous and synonymous mutations and dividing one by the other. This of course is not the case. Perhaps dN and dS could be described as rates or dN/dS as the dN:dS odds ratio which is how it's calculated for your permutation metric.

Line 112 – 40% of what, benefit of ~130% of what. This becomes apparent later into the manuscript but not clear how to interpret when reading at this point for the first time.

Line 188 – Mutational burden <= 3 (what units).

Line 189 – "We observed little negative selection in passengers" be clear what passengers (previous identified passenger genes).

Line 252 – Panel G, y-axis, what units? Why are "all" genes uniformly at approximately -0.2? Assuming this is fold-change or log-fold-change I'd expect 1 or zero respectively.

Line 275 – Figure 2G, shaded error bars are not visible.

---

## [Author Response]

Essential revisions:(1) The authors state that they are excluding tumors with either nonsynonymous(n)=0 or synonymous(s)=0 mutations. Since nonsynonymous and synonymous variants occur in a ratio of about 3:1, this exclusion of tumors would seem to lead to an inflation of the signal of selection in the first (lowest mutation) bins. Additional demonstration is required to show if this distorts the estimate of selection for low mutation burden tumours. For example, by adding pseudo-counts of mutations or by aggregation over tumours in the same mutation load "bin" as was performed for some analyses.

We thank the reviewers for bringing up this important point. Our original reasoning for applying this filtering procedure was the concern that false positive random mutations, which would push dN/dS to 1, would dominate the signal in tumors with very few true mutations. Nonetheless, we agree with the reviewers that this filtering procedure can introduce potential biases into dN/dS estimates and thus, we have now re-analyzed all the main and supplemental figures in the re-submission without this filtering step.

Overall, we find that although negative selection on passengers is still present in low TMB tumors after removing this filtering procedure, signals of negative selection are diminished (from dN/dS ~0.4 to ~0.7 when combining all tumors in TCGA and ICGC and using both null models of dN/dS).

**Author response image 1. sa2fig1:** 

Since we are no longer applying any quality control filters, we need to be particularly stringent about the mutational calls we use to ensure we don’t lose all signal to potential false positives. Indeed, we empirically observe that the power to detect selection is strongly dependent on the quality of mutation calls in the sample. We find that even within the same dataset (TCGA), signals of negative selection on passengers in low TMB tumors disappears (dN/dS ~ 1) when just one mutation caller is used (‘Mutect 2 SNP Calls’) but can still be detected (dN/dS ~ 0.5) when we use a consensus set of mutation calls (‘MC3 SNP Calls’; which uses 7 different mutation callers). This is true when using either null model of dN/dS (dNdScv and dNdS-permutation). Similarly, signals of positive selection on drivers in low TMB tumors also diminishes (from dN/dS ~5 to ~4) when low quality mutation calls are used. The figure is now included in the re-submission as Figure 2—figure supplement 15Since coverage in protein-coding regions in whole-genome data is much lower than in the whole-exome data and combining datasets can potentially introduce further bias, we elected to only use TCGA whole-exome data – which contains the most stringent consensus mutation calls (MC3 SNP Calls) – in the re-submission to focus on patterns of selection in the highest quality set of mutations available. We also note that others have similarly raised concerns about difficulties combining whole-genome sequencing and whole-exome tumors even using the same cancer samples, which produce only 75% of concordant mutations when comparing mutation calls in protein-coding regions (Bailey et al., 2020, Nature Communications). As expected, we find stronger signals of selection in drivers (from dN/dS ~5.5 to ~4) and passengers (from dN/dS in ~0.7 to ~0.6) in low TMB tumors (< 3 protein-coding mutations) in TCGA (‘MC3 SNP Calls’) when compared to the combining both TCGA and ICGC using both null models of mutagenesis.

(2) The decline of dN/dS on driver genes is the subject of the supplementary text and we note the efforts taken to disentangle Hill-Robertson interference effects from other possible explanations for dN/dS decay in drivers with increasing tumor mutation burden (TMB). Driver genes can be quite tissue-specific (and thus mus-identified for a tumor) and number of drivers per tumour estimated to span approximately 1 to 10 (Martincorena et al., 2017). Consequently, the fitting of a single set of model parameters and showing they do not match well the observed data (Supplementary note figure) is insufficient to exclude the misidentification of driver genes, or presence of nonsynonymous-neutral mutations in annotated driver genes, as an explanation for the decline in dN/dS with increased TMB. We think it important that a range of justifiable parameters are applied in this modelling, to test if the observed data is robustly outside reasonable parametrisation of the model.

Our goal for the modeling section was to investigate whether a simple evolutionary model of linkage with damaging passengers and advantageous drivers could reproduce similar patterns of selection observed in the empirical data. We agree with the reviewers that attenuation of dN/dS in drivers can be explained by processes other than Hill-Robertson interference. As the reviewers pointed out, we focus on alternative explanations for attenuation in the drivers within the supplemental portion of the text. In the previous submission, we explored a similar scenario that the reviewers suggest (i.e. 50% of neutral mutations accumulating in driver genes). However, we agree with the reviewers that this does not conclusively allow us to reject or make claims about the contribution of alternative models in drivers. To correct this, we have now included these potential alternative explanations suggested by the reviewers into the main text (Page. 11, Lines 16-26) and removed the supplemental note.

We would like to emphasize, however, that the focus of our paper is on negative selection on passengers. In all of the alternative models of dN/dS in drivers, passenger mutations are required to impose a fitness cost to be able to observe attenuation of negative selection in passengers.

(3) In many figures that show dN/dS as a function of n+s (starting with Figure 2A and extending to Figures S2, 3, 9, 10, 12, 22 and 25), there are no error bars indicated, as opposed to the statement in the figure caption. The error bars/shading should be shown. In Figure 2A, is the observed depletion in the second bin still significant?

To clarify, error bars were already displayed in all of the figures mentioned, but are currently visualized as shading (rather than traditional error bars). We recognize that the light opacity of this shading might make it difficult to visualize for some readers, especially in print. To correct this, we have now increased the opacity of the shading in all of the figures throughout the text so that this is clearer. We thank the reviewers for bringing up this concern.

After removing this filtering step in our analysis, we find that tumors in the second bin are no longer significant within Figure 2A.

Reviewer #1 (Recommendations for the authors):1. Figure panels should be called out sequentially. For example, Figure 2G is called out before Figure 2D. This happens throughout the text, including main and supplementary figures, and should be corrected.

We thank the reviewer for bringing up this point and have now corrected the text.

2. Figure 2G shows that mean gene expression of genes encoding chaperones and the proteasome increases with increasing mutational burden. What about protein abundance? Is this in agreement with gene expression?

We agree with the reviewer that this would be an interesting avenue to explore further. Unfortunately, the only proteomics data that currently exists within TCGA is RPPA, which is very limited (only ~100 genes have been profiled across tumors), and these particular gene sets of interest have not yet been assayed across tumors.

3. Figure 2 mentions error bars in the figure legend, but no panel displays error bars. This is also true for Figure S13 and other figures. Authors should display the error bars to which they are referring to make their analysis more convincing.

As mentioned above, error bars are already displayed in all of the figures the reviewer has mentioned but are currently visualized as shading (rather than error bars). We recognize that the light opacity of this shading might make it difficult to visualize for some readers, especially in print. To correct this, we have now increased the opacity of the shading in the figures so that this is clearer throughout the text. We thank the reviewer for bringing up this concern.

4. Pg. 9 line 295 describes results of the analysis across genes belonging to different GO terms. However, Figure S13 only shows 3 categories: chromosome segregation, transcription and translation. How were these categories chosen? What about other categories? Such cherry picking doesn't convincingly support the conclusions that no specific GO functions are enriched. Also, translational regulation shows higher dN/dS in low mutation tumors suggesting that there is positive selection for passengers in this category. Authors should discuss in their manuscript why this is the case.

We appreciate the reviewer’s concern that there was little justification for why these gene sets were chosen for analysis. These were chosen based on previous literature that has supported these groups of genes being relevant to protein misfolding, and thus we hypothesized they might be under constraint. We have now extended the figure captions to include this.

5. Figure S15 shows the attenuation in selection of CNAs across cancer subtypes and broad cancer groups. However, HNSC and kidney cancer appear to be the exceptions. Authors should provide an explanation for these observations in the main text.

There are many potential reasons for why this signal might be missing in certain tumor types or broad tumor groups. For example, tumors that are not copy number driven or tend to be on the low-spectrum of CNAs overall, might simply not have enough CNAs across all tumors to generate a strong enough signal.

6. Generally, copy number variations are considered to be > 50 bp. Is there a rationale as to why authors chose 100 kb to be their cut-off in Figure 2C? If the size of CNA is an important parameter, then authors should explain why that is.

100kb was used to partition copy number variation into ‘small’ and ‘large’ categories. Copy number variation <100kb was still considered in our study. As mentioned in the text, the reason we chose this cut-off is that the average length of a gene is ~100kb. Thus, we would expect the selective effects of CNAs that are sufficiently large to disrupt genes or multiple genes to be stronger.

7. Non-allelic recombination and non-homologous recombination mechanisms involving replication accidents that lead to chromosome breakage occur with some frequency in somatic cells. How does the frequency of these events impact the selection efficiency in cancer as it relates to drivers and passengers? Can this also be incorporated in their evolutionary model?

We agree that this would be an interesting phenomenon to model. However, it’s hard to make inferences on how this would impact the efficacy of selection overall in drivers and passengers. More empirical information about how frequently these events occur on average per generation in a cell is needed before one can accurately model these scenarios. Thus, we believe this is beyond the scope of the paper.

8. Authors mentioned that haploinsufficiency was not used in the model. What about loss of heterozygosity which is extensive in cancer genomes? Can this parameter be included in the evolutionary model and how would it impact the results?

Our goal for the modeling section was to present a simple model of Hill-Robertson interference to examine if it's plausible that this could explain patterns of attenuated selection with mutational burden as observed in the data. There are many attributes of cancer biology that were not considered and how the relaxation of certain assumptions might impact the results has already been documented in Supplementary File 1. In addition, this particular point about haploinsufficiency has already been explored by others (Lopez et.al 2020 Nature Genetics) and thus, was not considered here.

Reviewer #2 (Recommendations for the authors):1. The authors have taken great care to study single-nucleotide variants and large CNAs. It would be great if they could confirm their findings by also showing the effect on small insertions and deletions.

We agree with the reviewer that this would be an interesting direction to explore. However, we are not currently aware of any null models that take into account the expected behavior of neutral insertions/deletions. Thus, this is not something that we could explore in this manuscript.

2. Figure S5 is showing a bias in the determination of dN/dS from simulation results and the correlation between mutation rate and n+s. I am not sure I understand why dN/dS under a neutral simulation would be biased. Also, the low median correlation between n+s and the mutation rate (<0.4) is quite surprising. I would have expected these to be almost perfectly correlated. Likewise, I do not understand the formula after l. 631. It states that this is the joint density of the two Poisson random variables that denote nonsynonymous and synonymous mutation count, yet there is an additional unexplained factor in the denominator, which corresponds to the probability of s>0. If the simulation that underlies Figure S5 was also used in the ABC-based parameter inference, this would raise a serious cause for concern.

dN/dS in a neutral simulation could be biased because the probability of a nonsynonymous mutation is larger than the probability of a synonymous mutation (because of constraints of the genetic code). For low mutation counts, there are only certain integer totals possible (e.g. 2 nonsynonymous & 1 synonymous) which might subtly bias results. In the previous submission, Figure S5 showed that this bias is very subtle. For clarity, however, we have elected to remove this figure in the resubmission.

The equation on line 631 was not used in the ABC-based parameter inference, and it also does not explicitly consider any selection coefficient. The denominator of this equation depends on three variables dN dS and **λ**S, which are the observed nonsynonymous and synonymous mutation counts, and the mean number of synonymous mutations. The subscript S in these equations represents ‘synonymous’ mutations and not ‘selection coefficient’; we apologize for this overloaded use of nomenclature – these nomenclature choices predate our study. In the resubmission, this equation now explicitly states its parameters.

3. The simulation starts when the first mutation with positive selective effect initiates population growth, which can be very late in a patient's life. How does the assumption that up to 100 years can pass after this affect the parameter estimates?

To clarify, the model assumes that a tumor has equal probability of arising (via an initating driver or 1^st^ mutation with positive effect) at any point between 0 and 100 years of life. Thus, if a late-stage tumor arises (e.g. at age 70), the tumor doesn’t have 100 years from initiation (it has 30 years to become a successful tumor). This assumption has also been previously described in detail in (McFarland et al. 2014) and is common in mathematical models of cancer age-incidence rates. 4. To which extent does the inferred distribution of selection effects depend on the allowable parameter range? For example, s_passengers extends beyond the initially allowable range after the fit (Figure 3C).

The reviewed version of the manuscript had this figure reflecting ABC simulations extrapolating beyond the simulated range of parameters, which was needed to model the tail of the S_passenger probability distribution. This extrapolation may create unforeseen issues, so in the revised manuscript, we now simulate s_passengers from 10^-4^ to 10^-1^, and the new figures are updated accordingly. The posterior distribution of s_passengers did not change substantially; our estimate is now that S_passengers is 1.03% 95% CI [0.39%, 4.0%].

5. It is not entirely clear to me how the partitioning of the likelihood between Muller's ratchet and hitchhiking vs other effects can be made and how robust these inferences are with respect to variation of the modeling assumptions (e.g. about initial population size or mode of selection). Is the necessity of inclusion of selected synonymous variants on driver genes a robust result or not, taking into account the discussion on p. 26f.?

In the revised manuscript, we now indicate that the rates of Muller’s Ratchet and hitchhiking are derived from analytical theory in the Figure Legend and cite this theory. The robustness of these results are well-discussed in the papers that derive these formulae.

Reviewer #1 also raised concerned about explanations for the dN/dS attenuation of drivers, so we have removed the Supplementary Note discussing this topic.

6. In Figure S4, the authors report the correlation of n+s with other measures of tumor mutation load. Given the relative sizes of the different regions that are displayed, i.e. whole genome:intergenic:intronic:exonic:protein-coding, of roughly 100:60:40:2:1, the displayed numbers do not make sense, as their ratios are 100:100:100:1:0.001.

This figure has now been removed in the re-submission.7. I am not sure I understood well how CNAs were analyzed. Based on the description in l. 669ff., it appears that putative cancer driver genes were identified from the CNA data based on recurrence. Were the same data then analyzed for CNAs falling into said putative cancer driver gene regions to infer selection? This would appear a bit circular.

The copy number specific drivers were called using GISTIC 2.0, which are indeed based on recurrence across cancer types. We appreciate the reviewer’s point that the some of the data used to identify drivers was also used to infer selection on these genes. However, the metric used to infer selection (dE/dI) is a different method/static from the one GISTIC uses. Furthermore, this is a problem that generally applies to all methods that are currently used to infer drivers, even for point mutation-specific drivers.

8. I do not understand the formula shown after li. 738. It appears it is showing the fraction of genes that intersect a CNA boundary, summed over all tumors in a given n+s bin. Each CNA can be counted twice if both of its boundaries fall into a gene. Why is the mean value of this 1?

Within the methods section, we reference Figure 2—figure supplement 8 where we show that dEdI of random CNAs (where the start and stop locations are randomly permuted across the genome) recapitulates expected neutral values of 1.9. In all figures that show dN/dS as a function of n+s (starting with Figure 2A and extending to Figures S2, 3, 9, 10, 12, 22 and 25), there are no error bars indicated, as opposed to the statement in the figure caption. In Figure 2A, is the observed depletion in the second bin still significant?

As mentioned above, error bars are already displayed in all of the figures the reviewer has mentioned but are currently visualized as shading (rather than error bars). We recognize that the light opacity of this shading might make it difficult to visualize for some readers and have corrected the figures by increasing the opacity. The observed depletion of the second bin in passengers is no longer significant.

10. In l. 290, I understand that the authors argue that differential dominance effects between heterozygous early- and late-arising mutations could be affecting the efficacy of selection on subclonal variants compared to clonal variants. I do not see this claim well motivated or corroborated.

To clarify, we don’t claim that dominance affects the efficacy of selection. We were stating that early-arising mutations, which tend to be low frequency variants that are likely to be heterozygous, are expected to experience weaker selection overall.

11. In Figure 2D, the caption states that mutations have been separated into two groups by their clonality, yet the figure shows three curves. What do they correspond to? Are the results still significant given the partitioning of the mutation data into smaller subsets?

The three curves were meant for the reader to visualize and compare how selection in drivers and passengers changes when all variants are included (variants of all VAF values), only subclonal variants are included (VAF < 0.2) or only clonal variants are included (VAF > 0.2). In the resubmission, we have now removed variants of all VAF values to make it simpler to compare selection on clonal vs subclonal variants. The revised results in the resubmission are significant for clonal mutations, but not subclonal mutations.*12. Figure 3 does not have a panel G.*

This has now been corrected, thank you.

Reviewer #3 (Recommendations for the authors):I enjoyed reading the manuscript, it was well written, generally clear figures and very through provoking.

We thank the reviewer for these kind words.Only a small number of specific points to address:

Line 60.- The description of dN and dS here along with the interpretation of dN/dS=1 as neutral implies that you are just counting non-synonymous and synonymous mutations and dividing one by the other. This of course is not the case. Perhaps dN and dS could be described as rates or dN/dS as the dN:dS odds ratio which is how it's calculated for your permutation metric.

We appreciate the reviewer’s concern that the dN/dS calculations presented here are not simply counts of nonsynonymous and synonymous mutations. In the resubmission, we have now added the word rate in the main text to refer to our dN/dS calculations more accurately. However, we worry that adding additional terms to the metric ‘dN/dS’ can potentially add to more confusion – especially since others that calculate dN/dS with null models of mutagenesis simply use ‘dN/dS’ when referring to this statistic (Martincorena et al. 2017).

Line 112 – 40% of what, benefit of ~130% of what. This becomes apparent later into the manuscript but not clear how to interpret when reading at this point for the first time.

Thank you, we have now corrected this.

Line 188 – Mutational burden <= 3 (what units).

Thank you, we have corrected the text.Line 189 – "We observed little negative selection in passengers" be clear what passengers (previous identified passenger genes).

We have changed the text to clarify this.

Line 252 – Panel G, y-axis, what units? Why are "all" genes uniformly at approximately -0.2? Assuming this is fold-change or log-fold-change I'd expect 1 or zero respectively.

These values are z-scale normalized and this normalization was already performed by COSMIC when the data was downloaded.

Line 275 – Figure 2G, shaded error bars are not visible.

Thank you, this has been fixed.